# Cold-induced chromatin compaction and nuclear retention of clock mRNAs resets the circadian rhythm

Harry Fischl[1] (iD), David McManus[1,†] (iD), Roel Oldenkamp[1,‡] (iD), Lothar Schermelleh[1] (iD), Jane Mellor[1] (iD), Aarti Jagannath[2] (iD) & André Furger[1,*] (iD)

## Abstract

**Cooling patients to sub-physiological temperatures is an integral part of modern medicine. We show that cold exposure induces temperature-specific changes to the higher-order chromatin and gene expression profiles of human cells. These changes are particularly dramatic at 18°C, a temperature synonymous with that experienced by patients undergoing controlled deep hypothermia during surgery. Cells exposed to 18°C exhibit largely nuclear-restricted transcriptome changes. These include the nuclear accumulation of mRNAs encoding components of the negative limbs of the core circadian clock, most notably REV-ERBα. This response is accompanied by compaction of higher-order chromatin and hindrance of mRNPs from engaging nuclear pores. Rewarming reverses chromatin compaction and releases the transcripts into the cytoplasm, triggering a pulse of negative limb gene proteins that reset the circadian clock. We show that cold-induced upregulation of REV-ERBα is sufficient to trigger this reset. Our findings uncover principles of the cellular cold response that must be considered for current and future applications involving therapeutic deep hypothermia.**

**Keywords** chromatin compaction; circadian rhythm; cold stress; nuclear and cytoplasmic transcriptomes; REV-ERBα
**Subject Categories** Chromatin, Transcription & Genomics; RNA Biology
**The EMBO Journal (2020) 39: e105604**

See also: **RF Harvey & AE Willis** (November 2020)

## Introduction

Organisms are continually exposed to a variety of dynamic biological and environmental stressors. In order to survive, their cells must sense stresses and respond with structural alterations and the production of an adequate repertoire of metabolites and chaperones that enables them to reinstate cellular homeostasis and adapt to the new conditions (Kültz, 2020). Transcriptional reprograming (Vihervaara *et al*, 2018) and post-transcriptional (Chua *et al*, 2020) adjustments of specific mRNA and protein levels in response to a particular stress are central to this adaptation. One of the most common abiotic stressors organisms experience results from changes in their environmental temperature. These temperature fluctuations have a profound impact on cells, affecting molecular structures, enzyme activity, and the reactivity of metabolites, which can compromise viability.

We have a good understanding of how cells react to heat (Al-Fageeh & Smales, 2006; Vihervaara *et al*, 2018) but we know comparably very little about the gene networks and pathways that are activated when cells are exposed to cold (Ou *et al*, 2018). In bacteria, the gene expression response to cold stress is largely regulated at the post-transcriptional level where temperature-specific unfolding of mRNA structures by cold shock proteins (csps) regulates the global rate of translation initiation, elongation, and mRNA stability. At the heart of this regulation is the chaperone activity of the cold shock protein cspA and its 5′UTR that acts as a thermosensor (Giuliodori *et al*, 2010) and auto-regulates its own abundance in a temperature-dependent way (Zhang *et al*, 2018). In plants, cold adaptation is more complex, involving temperature-induced adjustments of the chromatin (Park *et al*, 2018; Zeng *et al*, 2019), transcriptional (Nagano *et al*, 2019), pre-mRNA processing (Calixto *et al*, 2018), post-transcriptional, and post-translational landscapes (Zhu, 2016). The transcriptional response to cold acclimatization in plants is beginning to be unraveled and entire signaling networks such as the C-repeat-binding transcription factor (CBFs) regulon have been exposed (Zhao *et al*, 2015). Our understanding of how animal cells, especially mammalian cells, respond and adapt to cold temperatures is minimal and almost entirely limited to temperatures above 28°C (Al-Fageeh & Smales, 2006). Cold exposure to these temperatures activates the expression of the highly conserved cold-inducible RNA-binding protein (CIRBP) and RNA-binding motif protein 3

1 Department of Biochemistry, University of Oxford, Oxford, UK
2 Sir William Dunn School of Pathology, University of Oxford, Oxford, UK
*Corresponding author. Tel: +44 (0) 1865 613 261; E-mail: andre.furger@bioch.ox.ac.uk
†Present address: MRC LMB, Cambridge Biomedical Campus, Cambridge, UK
‡Present address: The Netherlands Cancer Institute, Amsterdam, The Netherlands

(RBM3) (Zhu *et al*, 2016). These two RBPs govern mammalian cold stress adaptations by regulating pre-mRNA processing and translation of gene transcripts that encode proteins with protective, including anti-apoptotic properties (Zhu *et al*, 2016). The protective effects attributed to RBM3 and CIRBP are of considerable medical interest (Peretti *et al*, 2015; Bastide *et al*, 2017) and exemplify the therapeutic potential that understanding of cold response mechanisms bear.

Despite considerable efforts to date, how cold responsive proteins are activated and how they ultimately confer cellular protection is largely unclear (Bastide *et al*, 2017). The regulatory networks are highly complex and have recently been shown to extend as far as the cellular circadian clock. The circadian oscillations in temperature that organisms experience are sufficient to activate both RBM3 and CIRBP, which in turn can modulate circadian gene expression post-transcriptionally by interacting with core clock gene transcripts (Morf *et al*, 2012; Liu *et al*, 2013). Cellular clocks enable cells to align their physiology with daily environmental cycles by controlling the rhythmic expression of thousands of genes. At the heart of the mammalian cellular clock are two coupled cell autonomous transcriptional/translational feedback loops (TTFLs). The positive limb of both loops is formed by CLOCK and ARNTL (BMAL) binding as a heterodimeric complex to E-box elements in the promoters of periods 1 and 2 (*PER1*, *PER2*), cryptochromes 1 and 2 (*CRY1*, *CRY2*) and the nuclear receptor *REV-ERBα*, activating their transcription (Jagannath *et al*, 2017). In the negative limb of the main loop, CRY/PER proteins form a complex that, after reaching a threshold, acts as a co-repressor by binding to and inhibiting the activity of ARNTL/CLOCK, thus repressing *CRY/PER* transcription and that of *REV-ERBα*. In the negative limb of the stabilizing loop, accumulating REV-ERBα binds to the promoters of *ARNTL* and *CLOCK* to repress their transcription (Crumbley & Burris, 2011; Takahashi, 2017). These two tightly interlinked TTFLs result in rhythmic expression of these core clock genes, and, as ARNTL/CLOCK and REV-ERBα associate with numerous other promoters, thousands of clock-controlled genes. This creates a complex interwoven regulatory network that enables individual cells and organisms to synchronize cellular physiology and organismal behavior with the daily solar cycle (Koike *et al*, 2012).

The activation of both RBM3 and CIRBP, however, is restricted to a very narrow temperature range between 28 and 34°C (Rzechorzek *et al*, 2015). How mammalian cells respond to lower temperatures and how the activation of protective or detrimental pathways is controlled under such conditions is unknown (Ou *et al*, 2018). Work addressing cold responses in yeast (Kandror *et al*, 2004) and plants (Londo *et al*, 2018) suggest that different gene expression programs are activated depending on how low the temperature drops. While most cells in warm-blooded animals are maintained within a narrow temperature range, cells and tissues of the extremities (Brajkovic & Ducharme, 2006) and cells of small hibernating mammals (Ruf & Geiser, 2015) regularly sustain temperatures below 10°C. Furthermore, human cells in internal organs are exposed to below 30°C during a number of medical procedures. These approaches exploit the reduction in cellular metabolic needs associated with cooling to minimize tissue damage resulting from ischemia when blood supply to vital organs is disrupted (Quinones *et al*, 2014). Current medical applications of controlled extreme cooling include exposure of organs or whole patients to as low as 18°C for a range of cardiothoracic and neurological surgeries

conducted under deep hypothermic conditions (Mackensen *et al*, 2009). Transplant organs are routinely preserved by storage at temperatures below 10°C (Rao *et al*, 2001), and the body temperature of patients with major vascular damage are lowered to < 10°C during pioneering emergency resuscitation procedures (Kutcher *et al*, 2016). While the reduction of the metabolic needs of cells at very low (< 28°C) temperatures are widely recognized (Ou *et al*, 2018), there remains a major conceptual gap in our understanding of the mechanisms and pathways that govern beneficial and destructive gene network adaptations to these conditions. Understanding these mechanisms will be critical to fully exploit the therapeutic potential of controlled cooling (Kutcher *et al*, 2016).

Here, we aimed to address the conceptual deficit of the mammalian cold stress response by tracking cold-induced changes to human cardiomyocytes exposed to temperatures (28, 18 and 8°C) that are synonymous with the aforementioned medical procedures. We find that cold shapes both chromatin structures and subcellular transcriptomes in a temperature-specific manner. We discover that exposure to 18°C upregulates the transcript level of more than 1,000 genes, including the core circadian clock negative limb gene transcripts *PER1*, *PER2*, *CRY1*, *CRY2,* and *REV-ERBα*, but export of these transcripts into the cytoplasm is restricted. We provide evidence that the nuclear retention of the cold-induced transcripts at 18°C is caused by the compaction of chromatin into the nuclear periphery, compromising the approach of mRNPs to the nuclear pores. Export restrictions can be removed by rewarming cells back to 37°C, which reverses chromatin compaction and releases the accumulated nuclear transcripts into the cytoplasm. The sudden release of these mRNAs triggers a pulse of clock negative limb gene transcripts in the cytoplasm and their translation resets the circadian clock. Importantly, we show that REV-ERBα is the earliest to respond and most upregulated clock gene and its upregulation, following exposure to 18°C, is sufficient to reset the clock. Our findings expose a novel gene regulatory network and mechanism that links circadian gene expression with the mammalian response to very low temperatures.

## Results

### Cold exposure to 18°C triggers temperature-specific, reversible changes to the higher-order chromatin architecture

To understand how mammalian cells respond to temperatures below the normophysiological range (< 30°C) that cells of internal organs experience during a number of medical procedures, we exposed human cardiomyocyte AC16 cells (Davidson *et al*, 2005) to 28, 18, or 8°C. We then tracked temperature-dependent changes to the chromatin architecture by inspecting their 4′,6-diamidino-2-phenylindole (DAPI)-stained nuclei by super-resolution 3D-structured illumination microscopy (3D-SIM) (Schermelleh *et al*, 2008). This analysis revealed temperature-specific visible changes to the sponge-like fibrous structure formed by chromatin (Schermelleh *et al*, 2008; Figs 1A and EV1A–C, DAPI panels). To quantify these topological changes, we applied a segmentation algorithm (preprint: Miron *et al*, 2019) that assigns each voxel of the nuclear DAPI signal into chromatin-depleted interchromatin (IC) and chromatin-enriched regions (Fig 1A, segmented chromatin panels, Movies

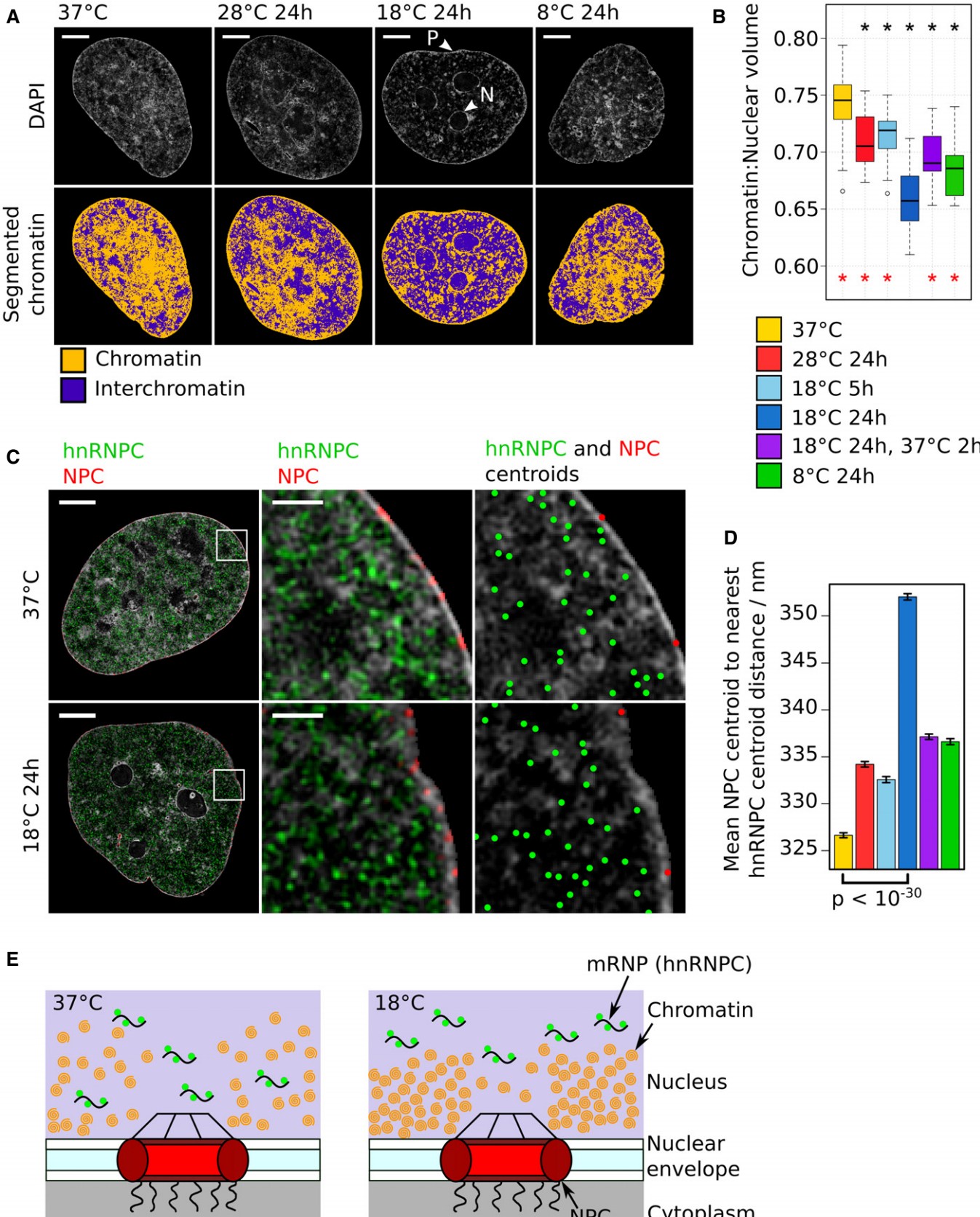

**Figure 1.**

◀

**Figure 1. Cold temperature induces changes to the higher-order chromatin architecture.**

A   Upper panels: Representative single z-planes from 3D-SIM image stacks of the DAPI-stained nuclei of AC16 cells kept at 37°C or exposed to different cold temperatures for 24 h. Lower panels: DAPI signal segmented into chromatin and interchromatin regions according to relative intensity. Scale bar: 5 μm. Arrows mark chromatin concentrated at the nuclear periphery (P) and around nucleoli (N).

B   Boxplots of the ratio of chromatin volume to nuclear volume for cells exposed to different temperature conditions. Median (central line), interquartile range (IQR, box edges), most extreme data point no more than 1.5× IQR from box edges (whiskers). Outliers outside the whiskers are shown as individual points. Asterisks mark mean ratios that are significantly [adjusted P-value < 0.05 (two-sided Mann–Whitney test)] different from the mean ratio of cells at 37°C (black asterisks) or cells exposed to 18°C for 24 h (red asterisks). Number of nuclei: 37°C (31), 28°C 24 h (26), 18°C 5 h (19), 18°C 24 h (22), 18°C 24 h then 37°C 2 h (25), 8°C 24 h (23).

C   Left and central panels: representative single z-planes from 3D-SIM image stacks of the DAPI-stained nuclei of cells kept at 37°C or exposed to 18°C for 24 h showing the spatial distribution of the immunofluorescence (IF) signal of hnRNPC (green) and nuclear pore complexes (NPCs, red). Right panel: DAPI signal overlaid with dots indicating IF signal centroid coordinates for centroids located in the z-plane shown. Scale bar whole nuclei: 5 μm. Scale bar enlarged section: 1 μm.

D   Bar graph of the mean NPC centroid to nearest hnRNPC centroid distance for each temperature condition. Error bars show the SEM. P-value (two-sided Welch's t-test) calculated for the comparison shown. Number of distances: 37°C (112,511), 28°C 24 h (94,521), 18°C 5 h (72,340), 18°C 24 h (74,998), 18°C 24 h then 37°C 2 h (92,896), 8°C 24 h (80,570).

E   Schematic illustrating how chromatin concentrated at the periphery after exposure to 18°C could widen the average NPC to nearest mRNP distance and thus hinder mRNA nuclear export.

Data information: Actual P-values are presented in the source data file.
Source data are available online for this figure.

EV1–EV8). As temperature does not affect the size of the nuclear volume (Fig EV1D), this reveals that at low temperatures chromatin undergoes compaction with a corresponding enlargement of the IC space. Surprisingly, this relationship is non-linear, as the greatest compaction and largest increase in IC occur at 18°C, not 8°C (Figs 1B and EV1E). Upon rewarming back to 37°C, this cold-induced compaction to chromatin reverses back toward levels seen in cells before cooling (Fig 1B). Given the relationship between cold stress and chromatin modifications in plants (Kim et al, 2015), we next explored the possibility that the observed compaction may be accompanied by large-scale changes to the epigenetic landscape that may in turn regulate cold exposure-mediated cellular reprograming. To assess this, we labeled the IC region as class 1 and further segmented the chromatin-enriched region into six classes from 2 to 7, denoting regions of increasing chromatin density. We then visualized the distribution of a number of chromatin features in cells exposed to the different cold temperatures across these seven chromatin density classes. These included serine 2-phosphorylated RNA polymerase II (Pol2S2P) and H3K4me3, features typically associated with actively transcribed chromatin, and H3K27me3 and H3K9me3 heterochromatin marks that can act as an impediment to cellular reprograming (Nicetto & Zaret, 2019). After 24 h at 28, 18, or 8°C, the heterochromatin marks remain enriched within the high-density chromatin classes 4–7 (Fig EV1A, B and F) and the active marks (Pol2S2P, H3K4me3) remain enriched within the low-density chromatin classes 2–3 (Fig EV1A, B and F). Thus, despite the global compaction of chromatin seen at low temperatures, including at 18°C, no cold-dependent large-scale changes in the epigenome topology or distribution of active transcriptional markers are evident in cold-exposed cells compared with cells kept at 37°C (Fig EV1F). This result suggests that the cold-induced chromatin compaction is mediated by physicochemical changes and is unlikely to be dependent on enzyme-driven modifications.

At 37°C, chromatin forms a diffuse network that is roughly evenly spread throughout the nucleus (Figs 1A and EV1A–C). In contrast, when chromatin compaction is at its greatest at 18°C, chromatin becomes visibly concentrated around nucleoli ("N", Fig 1A) and at the nuclear periphery ("P", Fig 1A), resulting in an apparent thickening of chromatin in these locations. To verify this thickening at the nuclear periphery, we visualized the location of nuclear pore complexes (NPCs) and hnRNPC molecules, which bind mRNAs in the IC space to form mRNA ribonucleoproteins (mRNPs) (Fig 1C). As these two features localize to opposing sides of the lamin-associated chromatin at the nuclear periphery, changes in the average distance from the center of each NPC focus (centroid) to its nearest hnRNPC centroid are representative of changes in the thickness of this peripheral chromatin. Chromatin compaction at 18°C correlates with an increase in this average NPC to hnRNPC centroid distance (Figs 1C and D, and EV1G–I). Upon rewarming, as chromatin at the periphery returns to a less compact state, there is a concomitant reduction in this average distance back toward levels seen in cells before cooling (Fig 1D). We conclude that 18°C cold exposure results in thickening of the chromatin at the nuclear periphery, which constrains the engagement of mRNPs with NPCs (Fig 1E), and predict that this reduces mRNA export rates.

## Cold exposure shapes subcellular transcriptomes in a temperature-specific manner

We next addressed whether cold exposure at the aforementioned temperatures and the resulting compaction of chromatin shapes the nuclear and cytoplasmic transcriptomes. To that end, we used a subcellular fractionation approach (Neve et al, 2016; Fischl et al, 2019) and sequenced the 3′ ends of nuclear and cytoplasmic polyadenylated RNAs from AC16 cells exposed to 28, 18, or 8°C for 24 h and compared their levels with those from cells kept at 37°C. Strikingly, each cold temperature results in its own specific effect on the subcellular transcriptomes (Fig 2A). At 28°C, transcript level changes in the nucleus are generally mirrored by a change in the same direction and to a similar extent in the cytoplasm (Figs 2B, top panel, EV3D and E) with 666 and 1,302 genes with transcript levels showing significant changes in the nucleus and the cytoplasm, respectively (Fig 2A). In stark contrast, at 18°C, while the transcript levels of over 2,400 genes are significantly altered in the nucleus, cytoplasmic transcript levels are largely unaffected (Fig 2A and B, middle panel). At 8°C, despite being the most extreme change in temperature, the nuclear transcriptome remains almost identical to that of cells kept at 37°C (Fig 2B, bottom panel). Fewer than 70 genes show significant changes in transcript levels, suggesting transcription and nuclear RNA degradation and export are suspended.

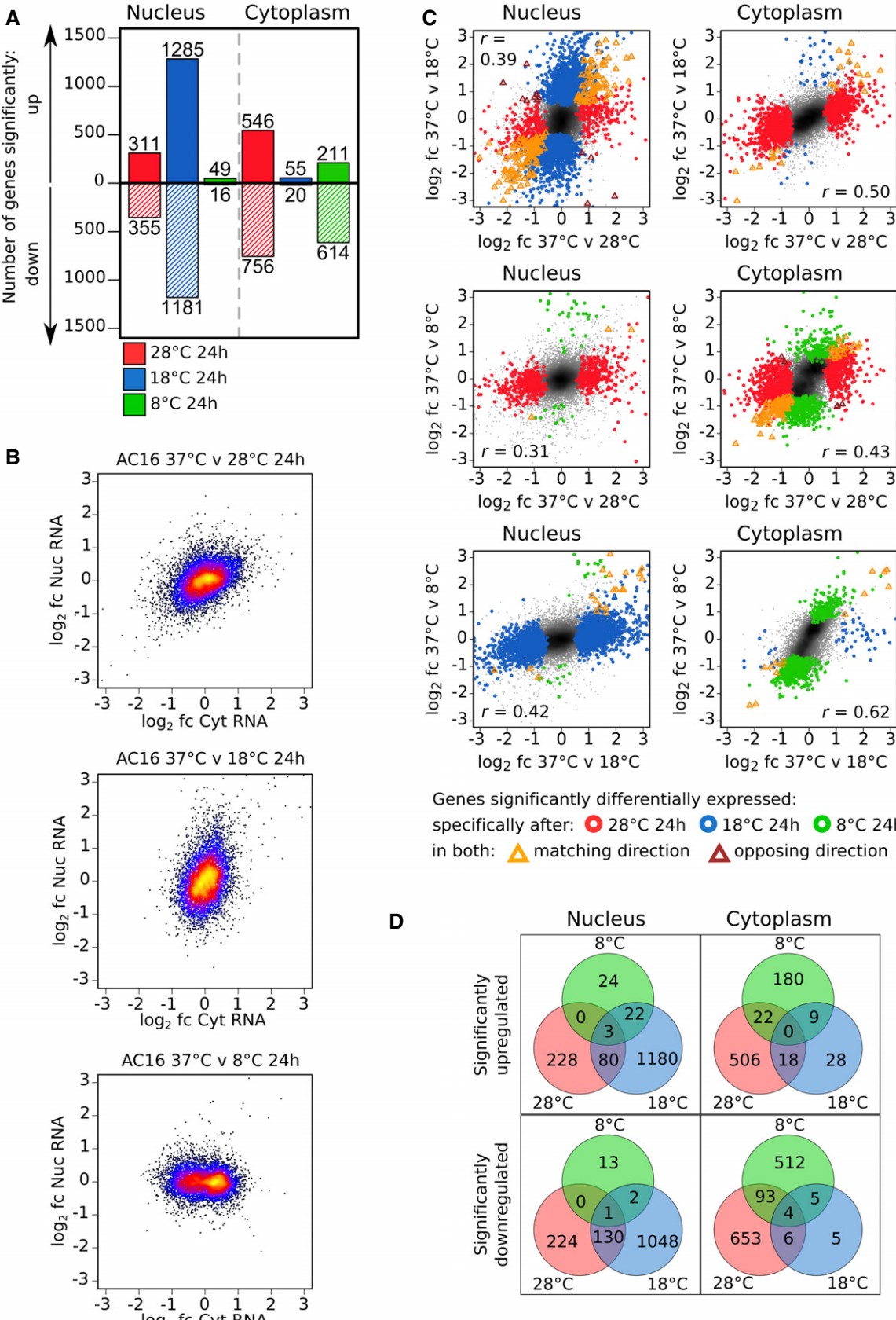

**Figure 2.**

**Figure 2. Cold induces temperature-specific changes to the nuclear and cytoplasmic transcriptome.**

A   Number of genes showing significant (adjusted *P* < 0.05, Wald test) up or downregulation in nuclear or cytoplasmic RNA levels upon transfer of AC16 cells from 37°C to different cold temperatures for 24 h.

B   Relationship between the $log_2$ fold change in RNA level in the nucleus and that in the cytoplasm for all genes upon transfer of AC16 cells from 37°C to different cold temperatures for 24 h. Points are color-coded from low to high density (black < blue < red < yellow).

C   Relationship between the $log_2$ fold change in RNA level for all genes upon transfer of AC16 cells from 37 to 28°C or 18°C (top), 28°C or 8°C (middle), and 18°C or 8°C (bottom) for 24 h in both the nucleus and cytoplasm. Points are shaded from low to high density (gray < black). Genes that show significant differential expression at a specific temperature or at both temperatures are highlighted with colored circles or triangles, respectively. Pearson correlation coefficient (*r*) is shown for each comparison (coefficient statistics are presented in the source data file).

D   Venn diagrams showing the overlap between the genes showing significant up or downregulation in nuclear or cytoplasmic RNA levels in AC16 cells exposed to the different cold temperatures.

Data information: Number of RNA-seq sample replicates: 37°C (N: 7, C: 6), 28°C 24 h (6), 18°C 24 h (6), and 8°C 24 h (6). Number of nuclear (N) and cytoplasmic (C) samples are the same for each condition except 37°C (see also Appendix Table S4).

Source data are available online for this figure.

In contrast, cytoplasmic RNA levels do show changes at 8°C (Fig 2B) with a significant reduction for more than 600 genes (Fig 2A). Those showing greater decreases tend to have shorter half-lives (Lugowski *et al*, 2018; Fig EV2A), suggesting cytoplasmic RNA degradation mechanisms are still active at 8°C. Transcriptome changes occurring in response to each cold temperature and within each compartment show only a weak positive correlation with those occurring in response to the other cold temperatures (Fig 2C). Importantly, there is minimal overlap between the genes showing significant changes between the different temperatures (Fig 2D). This is true for the known cold-inducible gene *CIRBP* transcript level, which is upregulated at 28°C but unaffected at 18 and 8°C (Appendix Table S1, Fig EV2B). This confirms that it plays no part in the cellular adaptation process to very low temperatures. The *RBM3* transcript level, which has also been shown to play a role in cold adaptation, is not upregulated at any of these cold temperatures (Fig EV2B). From these results, we conclude that the gene expression response to cold in mammalian cells is temperature and compartment-specific with little overlap between the gene networks that are activated at the different temperatures.

## Upregulated transcripts at 18°C are nuclear-retained and are released into the cytoplasm by rewarming to 37°C

Emulating its impact on chromatin architecture, exposure to 18°C also has the most dramatic impact on the transcriptome. While over 1,200 genes show upregulation of their nuclear RNA levels, their cytoplasmic levels remain unaffected (Fig 2A and B). Importantly, spike-in controls confirm that these changes represent changes in absolute levels (Fig EV2C and D). The restriction of accumulated mRNAs to the nucleus suggests that while transcription is still active and responding to the 18°C cold exposure, nuclear-cytoplasmic mRNA export is compromised at this temperature. Furthermore, the disconnect between nuclear and cytoplasmic expression changes gradually intensifies and includes a growing number of genes the longer the cells are exposed to 18°C (Fig EV2F–H). Interestingly, rewarming cells back from 18 to 37°C leads to a rapid readjustment of the nuclear-accumulated transcripts back to levels close to those seen before cooling (Fig 3A, top panel). This is concomitant with an increase of these transcripts in the cytoplasm (Fig 3A, bottom panel). Likewise, the 1,181 downregulated genes in the nucleus at 18°C are rapidly readjusted back to levels close to those seen before cooling after rewarming (Fig EV3A). Moreover, these changes are not specific to AC16 cells, as the nuclear and cytoplasmic

transcriptomes of U2OS cells, exposed to 18°C and then rewarmed, change in a similar way (Figs EV2E, and EV3B and C). This type of response is specific to 18°C as in cells that are exposed to 28°C, nuclear and cytoplasmic changes are mirrored during the entire cooling and rewarming cycle (Fig EV3D and E). Interestingly, the release of nuclear-accumulated transcripts into the cytoplasm after rewarming from 18°C coincides with the relaxing of the cold-induced compaction to chromatin (Fig 1B), providing evidence that these events may be linked (Fig 1E).

## Cold exposure to 18°C triggers nuclear-restricted upregulation of core clock negative limb genes

We next explored the physiological consequence of this sudden release of nuclear-accumulated transcripts into the cytoplasm after rewarming cells from 18°C. To identify potential impacts, we subjected the cohort of transcripts that are upregulated in the cytoplasm after rewarming to gene ontology (GO) analysis, revealing an enrichment for genes associated with "circadian rhythm" (Appendix Table S2). Cellular circadian rhythmicity enables anticipatory alignment of physiological processes with daily environmental cycles with wide-ranging implications for health (Jagannath *et al*, 2017; Montaigne *et al*, 2018). As outlined earlier, these rhythms are orchestrated by a self-sustaining, cell autonomous, molecular clock consisting of transcriptional activators of the positive limb, including ARNTL and CLOCK, and corepressors/repressors or the two negative limbs, including PER1/2, CRY1/2, and REV-ERBα, which interact in auto-regulatory feedback loops to generate oscillations in their expression levels that typically have 24 h periodicity and distinct phases (Takahashi, 2017).

To explore the impact of the 18°C cold exposure on core clock gene expression in more detail, we tracked the nuclear and cytoplasmic mRNA levels of these genes individually during exposure to 18°C and subsequent rewarming in AC16 (Figs 3B and EV3F) and U2OS cells (Fig EV3G). While the activators, *ARNTL* and *CLOCK*, do not show significant changes, negative limb components of the clock show nuclear-restricted accumulation of fully processed mRNAs (Fig 3C). There are subtle differences in how the negative limb genes respond to 18°C. *REV-ERBα* accumulates rapidly in the nucleus after 5-h exposure to 18°C and after 24 h also seeps into the cytoplasm (Fig 3B). *CRY1, CRY2,* and *PER1* increase after 24 h at 18°C but their increase is nuclear-restricted (Fig 3B). The nuclear-retained transcripts surge into the cytoplasm after rewarming to 37°C where, as shown for *REV-ERBα*, they can be translated

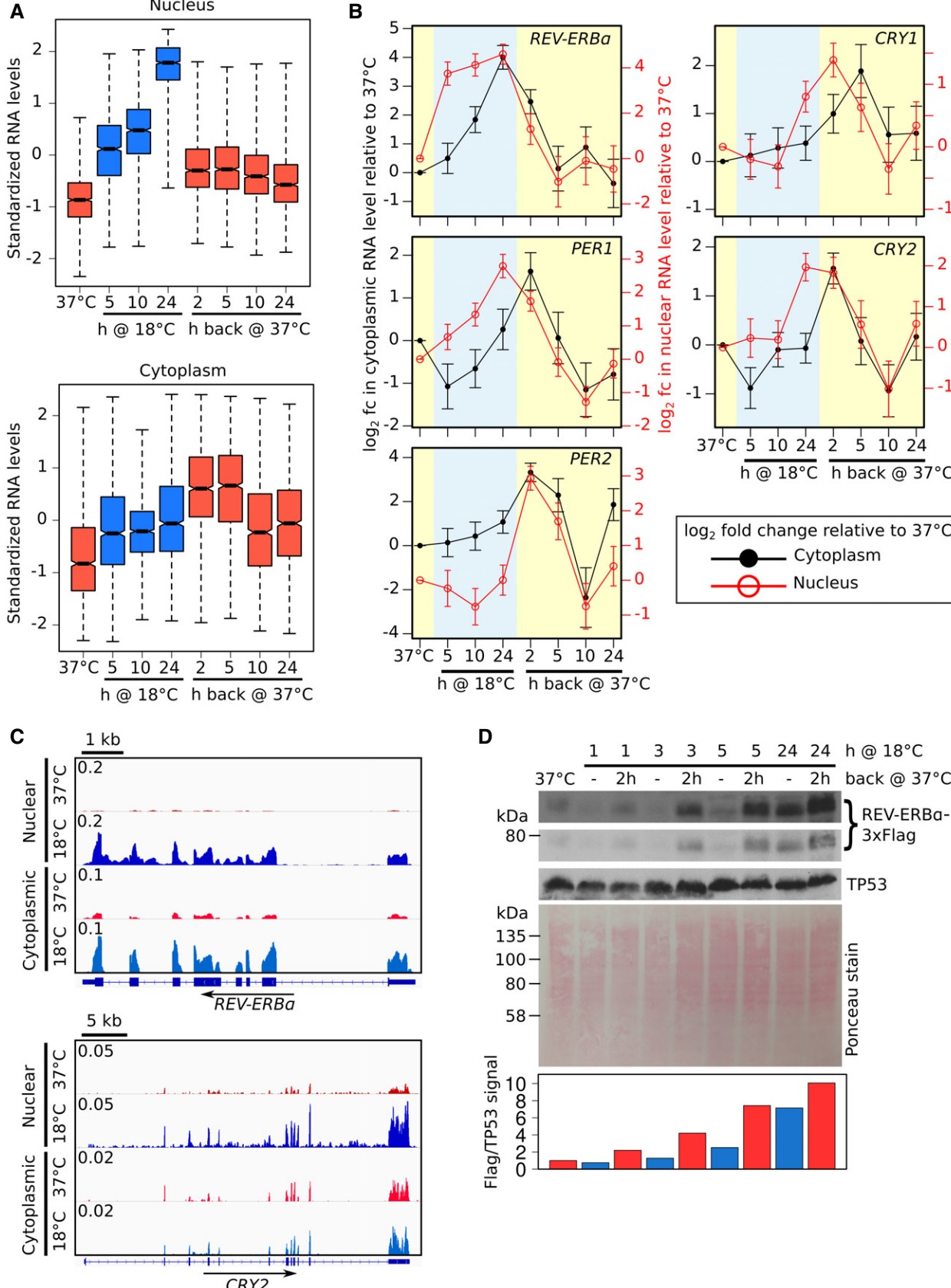

**Figure 3.**

◀

Figure 3.   Rewarming following 18°C exposure releases nuclear-accumulated transcripts into the cytoplasm, including those of core circadian clock negative limb genes.

A   Boxplots of standardized nuclear and cytoplasmic RNA levels at time points during the transfer of AC16 cells from 37 to 18°C for 24 h and then back to 37°C for 24 h for the group of 1,285 genes showing significant upregulation in nuclear RNA levels at the 18°C for 24 h time point. Range (whiskers), interquartile range (IQR, box edges), median ± IQR/[square root(number of genes)] (notch) and median (central line).

B   Log$_2$ fold change in cytoplasmic (black line, left axis) and nuclear (red line, right axis) RNA level of core circadian clock negative limb genes at time points during the transfer of AC16 cells from 37 to 18°C for 24 h and then back to 37°C for 24 h relative to cells kept at 37°C. Error bars show the standard error.

C   Nuclear and cytoplasmic full-length RNA-Seq data from AC16 cells kept at 37°C or exposed to 18°C for 24 h for REV-ERBα and CRY2. Data for each track have been normalized to show counts per million aligned counts (CPM) for each nucleotide. Panels were prepared using Integrative Genomics Viewer (IGV).

D   Western blot of Flag-tagged REV-ERBα levels (short and long exposure) in whole cell extracts from AC16 cells transferred from 37 to 18°C for the time periods indicated and also returned after each of these time periods to 37°C for 2 h. TP53 was used as a loading control as its transcript levels show minimal changes in response to 18°C exposure (Figs 4C and G, and EV3F). Ponceau stain is shown as an additional loading control. Quantification of the REV-ERBα/TP53 signal is shown below.

Data information: Number of 3′ end RNA-seq sample replicates: 37°C (C: 6, N: 7), 18°C 5 h (4), 10 h (4), 24 h (6), 18°C 24 h then 37°C 2 h (4), 5 h (2), 10 h (2), and 24 h (2). Number of cytoplasmic (C) and nuclear (N) samples are the same for each condition except 37°C (see also Appendix Table S4). Additional statistical details are presented in the source data file.

Source data are available online for this figure.

(Figs 3D and EV3H). Unlike the other clock negative limb genes, PER2 mRNA levels increase both in the nucleus and cytoplasm only after rewarming (Figs 3B and EV3G, Appendix Fig S1). This suggests that PER2 is activated by the rewarming process perhaps emulating a heat-shock response. However, it is unlikely that in our system rewarming induces PER2 expression via the HSF1 pathway, as previously reported (Kornmann et al, 2007; Buhr et al, 2010; Tamaru et al, 2011), because we found no evidence that heat-shock genes are activated during the cooling or subsequent rewarming process (Appendix Table S1).

To corroborate the observed RNA-Seq expression patterns of clock genes, we employed RNA-FISH using probes targeting the following transcripts: REV-ERBα, showing the highest upregulation and earliest response (Fig 3B); CRY2, showing a later response and nuclear-restricted upregulation (Fig 3B); TP53, showing no changes and acting as a control (Fig EV3F). This analysis confirms that, after 5 h at 18°C, REV-ERBα transcripts show rapid nuclear-restricted upregulation in both AC16 (Fig 4A and E) and U2OS (Fig 4D and H) cells, that, after 24 h, CRY2 transcripts accumulate in the nucleus and are released into the cytoplasm after rewarming (Fig 4B and F), and that TP53 transcript levels remain largely unchanged during cooling and rewarming (Fig 4C and G).

Collectively, these results show that the nuclear-restricted upregulation of the core clock negative limb gene transcripts is characteristic of the response to 18°C exposure and upon rewarming this nuclear pool of clock negative limb transcripts surges into the cytoplasm.

## Cold-induced upregulation of clock negative limb gene expression changes the phase and amplitude of the cellular circadian rhythm

We next investigated how this sudden burst of clock negative limb gene expression following rewarming affects the cellular circadian rhythm. In particular, we focused on the role of REV-ERBα given that, out of the clock gene transcripts, it is the only one to show upregulation in the nucleus after 5 h, while also showing the greatest upregulation after 24 h at 18°C (Fig 3B). To monitor cellular circadian oscillations in real time, we tracked the bioluminescence generated by a PER2 promoter-controlled luciferase reporter gene (PER2::LUC) stably expressed in U2OS cells with (WT cells) and

without REV-ERBα (KO cells) (Fig EV4A). In agreement with previous studies (Zhao et al, 2016), for cells kept at 37°C, REV-ERBα deletion increases the amplitude and period length of the circadian rhythm (Fig EV4B and C). We synchronized the cells with dexamethasone at regular intervals to produce cells with six distinct oscillating bioluminescence profiles distributed across the circadian period prior to cold exposure (six differently colored lines, Fig 5A, −48 to 0 h time window). This enables us to assess the relationship between the position of the cells within the circadian period at the point of cold exposure and any effect that the cold has on the subsequent amplitude and phase of the circadian rhythm. We first tested the effect of exposing cells to 18°C for 24 h and subsequent rewarming, which results in elevated expression of PER1/2, CRY1/2, and REV-ERBα (Fig 3B). For all profiles, this causes a large significant increase in amplitude in both WT and KO cells (Figs 5A and EV4D, 24–96 h time window) compared with the amplitude of the profiles of cells kept at 37°C (Figs EV4H and EV5C). In addition, it resets the clock by forcing cells into the same phase (clustered colored lines, Figs 5A and EV4D, 24–96 h time window) irrespective of the phase at which they oscillated before exposure to cold (Figs 5A, and EV4D, H and EV5B). To quantify this, for each profile we measured the time from the start of cold exposure to the second peak after rewarming and compared these with the time to the corresponding peak for profiles from cells kept at 37°C (peak "x", Fig 5C). The variance of these times for the different profiles is highly significantly reduced following exposure to 18°C for 24 h, confirming that the phases have shifted, resulting in more closely aligned profiles (Fig 5D). As this effect is independent of REV-ERBα levels (Fig 5D, top panel: WT to KO cell comparison), we propose that the increased expression of the other clock negative limb genes is sufficient to reset the phase, masking any potential role for REV-ERBα.

Rewarming following exposure to 18°C for 5 h triggers a cytoplasmic pulse of increased REV-ERBα without affecting PER1 or CRY1/2 levels (Figs 3 and 4, Appendix Fig S1). We could therefore use this to assess the effect of this cold-induced cytoplasmic surge of REV-ERBα on the circadian rhythm in isolation from the other clock negative limb genes, except for PER2, which shows increased transcript levels after rewarming (Appendix Fig S1). Upon rewarming following this shorter exposure, the phases of the WT cell profiles again become highly significantly more closely aligned than for cells kept at 37°C (Fig 5B, left panel, 5D, bottom panel). Strikingly, this

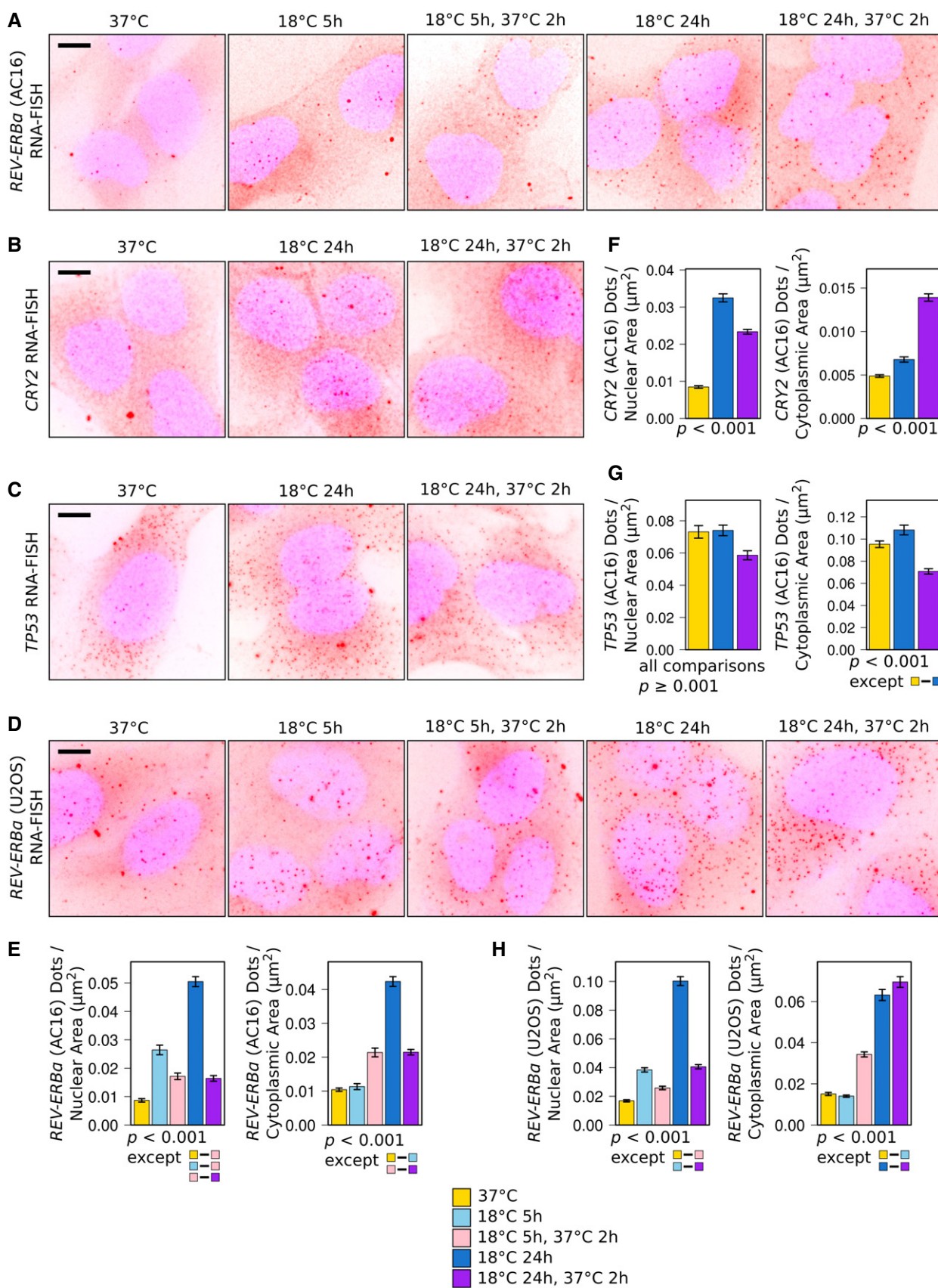

**Figure 4.**

◄

**Figure 4.  Nuclear accumulation upon 18°C exposure and release upon rewarming of core circadian clock negative limb gene transcripts.**

A–D   Representative maximum intensity z-stack-projected images of DAPI-stained (blue) AC16 (A–C) or U2OS cells (D), exposed to 18°C and returned to 37°C for the time
       periods indicated, probed by RNA-FISH for *REV-ERBα* (A and D), *CRY2* (B), or *TP53* (C) transcripts (red dots). Scale bar: 10 μm.

E–H   Mean *REV-ERBα* (E and H), *CRY2* (F), or *TP53* (G) transcripts per AC16 (E–G) or U2OS (H) cell nuclear and cytoplasmic area across all images for each condition from
       quantification of the RNA-FISH data. $P < 0.001$ (Tukey's HSD) for all pairwise comparisons except those shown. All error bars (E–H) show SEM.

Data information: Number of images: AC16 cells (*REV-ERBα*): 37°C (65), 18°C 5 h (30), 18°C 5 h then 37°C 2 h (21), 18°C 24 h (63), 18°C 24 h then 37°C 2 h (64); AC16 cells
(*CRY2*): 37°C (43), 18°C 24 h (42), 18°C 24 h then 37°C 2 h (42); AC16 cells (*TP53*): 37°C (47), 18°C 24 h (45), 18°C 24 h then 37°C 2 h (48); U2OS cells (*REV-ERBα*): 37°C
(61), 18°C 5 h (61), 18°C 5 h then 37°C 2 h (69), 18°C 24 h (53), 18°C 24 h then 37°C 2 h (65) (see also Appendix Table S7). Counts per image and actual *P*-values are
presented in the source data file.

Source data are available online for this figure.

alignment now requires REV-ERBα expression as the profiles of KO cells do not align after cold exposure (Fig 5B, right panel, 5D, bottom panel: WT to KO cell comparison; Fig EV4E and I; four additional profiles, EV4F, G and J). To dissect the impact of elevated REV-ERBα levels in relation to its own rhythmicity, we calculated the phase and amplitude change for each profile (Figs 5E, and EV5A, E and F) and plotted these in relation to the position of the cells in the circadian rhythm at the point of rewarming (Fig EV5D). This reveals that the REV-ERBα pulse forces a large significant phase delay when REV-ERBα (Fig EV5E, gray curve) should be decreasing (Fig EV5E: profiles d,y,e,f,z). This is greater for the profiles where REV-ERBα expression is nearing its minimum (Fig EV5E: profiles e, f,z), which also show large significant fold decreases in amplitude (Fig 5E: profiles e,f; Fig EV5F: profile z). This may be because the minimum REV-ERBα level reached may no longer be as low due to the cold-induced pulse. Conversely, if the cold-induced pulse of REV-ERBα expression occurs when REV-ERBα should be increasing, there is a large significant fold increase in amplitude (Fig 5E, profiles a,b,c; Fig EV5F: profiles w,x). This is likely because the maximum REV-ERBα level reached may now be higher. These results show that a single cold exposure to 18°C and subsequent rewarming creates a pulse in the expression of the core clock negative limb genes that changes the amplitude and resets the phase of the circadian rhythm. Importantly, cold-induced activation of REV-ERBα is sufficient to reset the phase.

Our results demonstrate that the core clock negative limb genes are part of a specific cellular response to very low temperature exposure that triggers resetting of the circadian rhythm upon rewarming.

# Discussion

Controlled cooling is an integral part of many clinical procedures, principally aiming to reduce the cellular metabolic needs of organs and tissues to prevent ischemic cell death when blood supply is interrupted. While the benefit of reduced metabolic rates for cell survival has long been appreciated, the activation of specific pathways has only recently been recognized as a critical contributor to the cell preservation effect afforded by low temperatures (Peretti *et al*, 2015; Bastide *et al*, 2017; Ou *et al*, 2018; Jackson & Kochanek, 2019). Unlike the metabolic rate that gradually reduces as temperature decreases, we show that the activation of pathways in response to cold is more complex. The mammalian response to cold is not scaled to temperature (Figs 1 and 2) and is finely balanced between beneficial and detrimental outcomes (Hattori *et al*, 2017), limiting its therapeutic scope. The importance of controlled cooling as a clinical tool (Gordon, 2001; Peretti *et al*, 2015; Kutcher *et al*, 2016;

Bastide *et al*, 2017) and its potential to provide solutions to overcome the physiological challenges of long duration space flight (Choukèr *et al*, 2019; Jackson & Kochanek, 2019; Nordeen & Martin, 2019) expose the need to understand the cellular response to cooling at the molecular level. Elucidating these processes will ultimately provide solutions to pharmacologically activate beneficial pathways without the need for cooling and inactivate detrimental pathways to expand the remit of controlled cooling for medical applications.

Here, we describe a multipronged approach to elucidate how cells respond to cold temperatures that are currently applied in a number of clinical settings. Using this approach, we find that, similarly to plants and yeast, human cells respond to cold in a highly temperature-dependent manner and that the activation of core clock genes is an integral part of the mammalian response to extreme cold temperature exposure.

The temperature-dependent cellular response is most apparent in the dramatic compaction of chromatin in cells exposed to cold, particularly at 18°C (Fig 1). Surprisingly, our 3D-SIM approach shows that the degree of compaction is non-linear and independent of enzymatic activity and is thus most likely caused by a physico-chemical change in the nucleus. A probable trigger for this change is a temperature-induced influx of calcium that is stored in the nuclear envelope and the nuclear reticulum (Bootman *et al*, 2009). Not only is calcium known to boost compaction of chromatin fibers (Phengchat *et al*, 2016) but calcium release channels are activated in a temperature-dependent manner, operating as thermo-sensors in both mammalian (Bautista *et al*, 2007) and plant cells (Ding *et al*, 2019). As both the calcium channel activity and membrane fluidity (Al-Fageeh & Smales, 2006) vary at different temperatures, the calcium release is likely to differ at the three temperatures, resulting in the observed differences in the degree of compaction.

The use of the nuclear-cytoplasmic fractionation approach added a novel dimension to the mammalian cold response by identifying the establishment of distinct temperature-specific nuclear and cytoplasmic transcriptomes (Fig 2). These differences are due to temperature-specific effects on mechanisms controlling transcription, mRNA export, and mRNA degradation. The activation of specific programs depends on how low the temperature drops, resulting in gene expression outputs with very little overlap between the affected genes at the different temperatures (Fig 2C and D). At 28°C, changes compared with cells kept at 37°C are characterized by the upregulation of the transcript level of around 500 genes, including the known cold-induced gene *CIRBP*. The changes in the nuclear and cytoplasmic transcriptomes are generally matched, suggesting that the predominant response at 28°C is at the transcriptional level. This is in marked contrast to 8°C where the nuclear transcriptome is indistinguishable from that of cells kept at 37°C. This may be caused by a depletion of

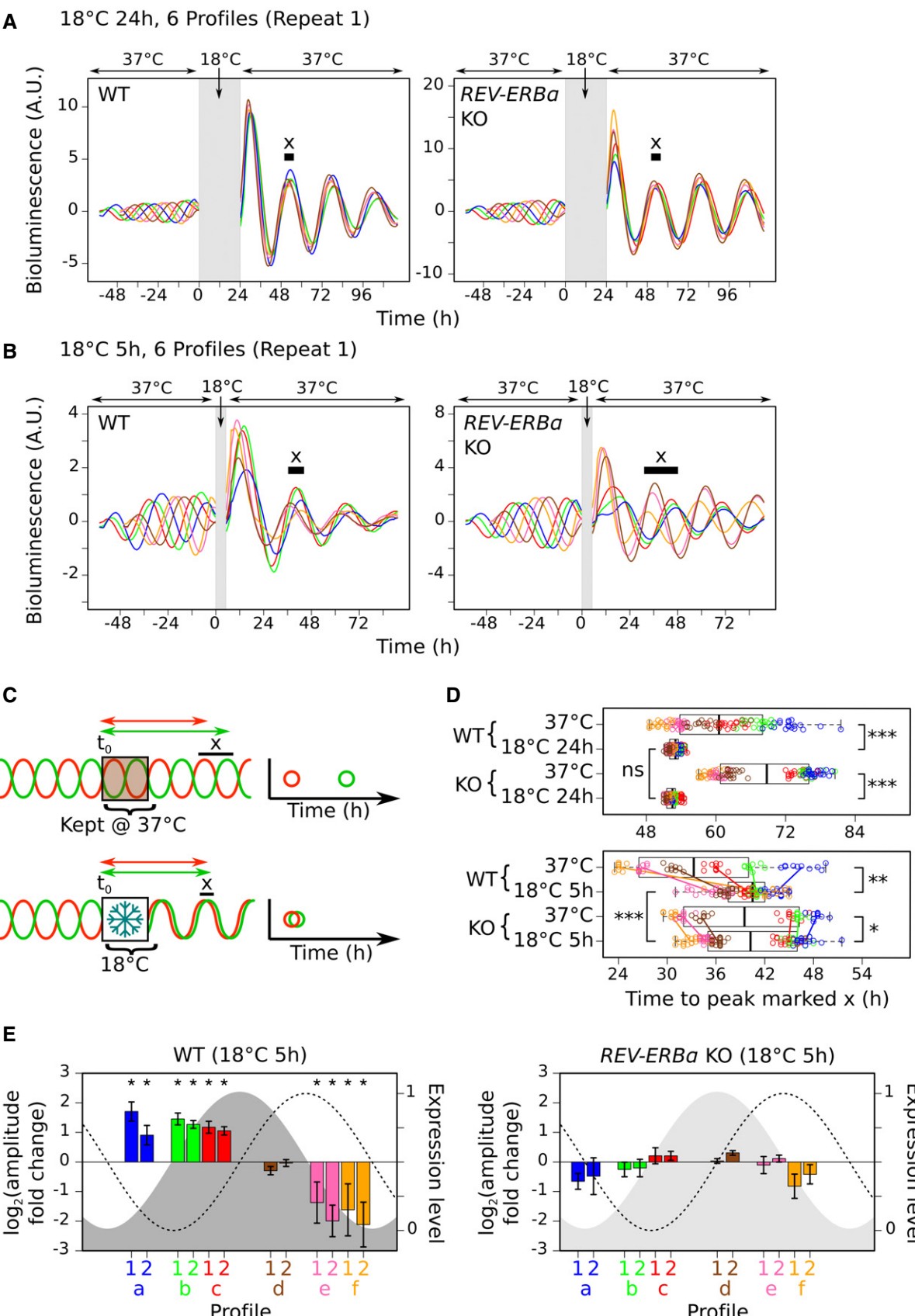

**Figure 5.**

Figure 5. Cold-induced REV-ERBα expression resets the phase and modulates the amplitude of the circadian rhythm.

A, B    Mean baseline-detrended bioluminescence profiles from plate wells containing either PER2::LUC U2OS cells with (WT, left panel) or without REV-ERBα (KO, right panel) recorded at 37°C before and after transfer to 18°C for 24 h (A) or 5 h (B) (gray region) from biological repeat 1. 6 differently colored profiles represent cells synchronized at 6 distinct phases of the circadian period prior to the start of 18°C exposure (time zero). Peaks marked "x" defined in (C).

C    Schematic showing the measurement of each profile from time zero to the peak marked "x" [this is the second peak after 18°C exposure or the corresponding peak in cells kept at 37°C (see (A, B) and Fig EV4D–F and H–J)]. Plotting these times shows the effect of 18°C exposure on the spread of these profiles.

D    Boxplots and individual points for times measured as in (C) for each plate well profile for both WT and KO cells kept at 37°C or transferred to 18°C for 24 h (top panel) or 5 h (bottom panel) from biological repeats 1 and 2. Points are colored according to their distinct phase prior to 18°C exposure. Colored lines (bottom panel) show the change in the mean for the points from each phase. Pairwise comparisons test the significance of the difference in variance (Brown–Forsythe test (adjusted for multiple testing)) ($^{ns}P > 0.05$; $*1 \times 10^{-10} < P < 0.05$; $**1 \times 10^{-20} < P < 1 \times 10^{-10}$; $***P < 1 \times 10^{-20}$). Boxplots display median (central line), interquartile range (IQR, box edges), and range (whiskers). Number of wells: top panel WT cells: 37°C (120), 18°C 24 h (119); top panel KO cells: 37°C (120), 18°C 24 h (120); bottom panel WT cells: 37°C (58), 18°C 5 h (120); top panel KO cells: 37°C (60), 18°C 5 h (120).

E    Bar graphs showing the mean amplitude change (left axis) following 18°C exposure for 5 h for each group of plate well profiles (grouped according to their distinct phase (a–f) prior to 18°C exposure, 10 wells per group) for both WT and KO cells from biological repeats 1 and 2. Mean changes are plotted to approximately align with the position of the control profile at the point of rewarming within the circadian period that is represented by the PER2:LUC (dashed line) and predicted REV-ERBα expression level (right axis) (gray shaded region (light gray indicates REV-ERBα deletion)). Error bars show the standard deviation. Asterisks mark mean $\log_2$ amplitude fold changes significantly greater than $\pm \log_2(1.4)$ (adjusted $P < 0.05$, one sample $t$-test). [see Appendix Fig S2 (Fig 5 extended data for more details)].

Data information: Actual P-values are presented in the source data file.
Source data are available online for this figure.

ATP at this temperature which will impair transcriptional rates (Pires et al, 2010), reduce the free movement of mRNPs away from chromatin, and prevent their nuclear export (Vargas et al, 2005), essentially "freezing" the status quo in the nucleus. Furthermore, it is likely that RNA polymerase II enzyme activity is much more severely compromised at 8°C compared with 18°C where a transcriptional response is still maintained. However, it is worth noting that the transcriptional response at 18°C is gradual and requires exposure for 24 h to affect the transcript levels of 1,000 genes (Fig EV2F). In contrast to the transcriptional activity, cytoplasmic degradation pathways are still active at 8°C leading to greater depletion of unstable transcripts (Fig EV2A), causing the disparity between nuclear and cytoplasmic 8°C transcriptomes (Fig 2).

The most dramatic transcriptome change is associated with shifting cells to 18°C, which causes a striking upregulation and retention of over 1,000 gene transcripts within the nucleus (Figs 2–4). Incomplete pre-mRNA processing has been identified as a key determinant for nuclear transcript retention (Yin et al, 2020), but this mechanism is unlikely to be responsible for the cold-induced restriction to export because the upregulated retained transcripts are polyadenylated and no evidence for increased intron retention was observed (Fig 3C). Furthermore, the surge of the upregulated transcripts into the cytoplasm and their translation (Fig 3D) after rewarming confirms that these transcripts are fully matured. Our 3D-SIM and RNA-FISH data suggest that the nuclear accumulation is linked to the high degree of chromatin compaction at 18°C. We propose that the cold-induced thickening of peripheral chromatin hinders mRNPs from approaching the nuclear pores along the chromatin free channels (Schermelleh et al, 2008) and that the compaction of the lamin-associated chromatin together with the low temperature affects the rigidity of the membrane, altering the mechanical properties of the NPCs. Collectively, these effects result in almost complete inhibition of nuclear-cytoplasmic mRNA export, creating the highly polarized subcellular transcriptomes characteristic for this temperature (Fig 1E). Importantly, these effects are reversible and rewarming prompts relaxation of the chromatin and release of the accumulated transcripts into the cytoplasm.

Interestingly, rewarming cells from 18°C rapidly revert the chromatin configurations and readjusts both the nuclear and cytoplasmic transcriptomes toward precooling states (Figs 3 and 4). With the

exception of a few gene transcripts, this process takes less than 2 h for the nuclear transcriptome and less than 10 h for the cytoplasmic transcriptome (Fig 3A), suggesting a high degree of coordination between mechanisms that control the rates of transcription and degradation during this short period.

A key finding of our detailed transcriptome analysis is the recognition that the core clock negative limb genes of both TTFLs (PER1, CRY1/2 and REV-ERBα) can be classified as cold response genes. The nuclear-restricted upregulation of the transcripts of these genes is dramatic, most notably for REV-ERBα, which exceeds a greater than 10-fold change in transcript abundance at 18°C compared with levels in cells kept at 37°C (Fig 3B). Importantly, our data demonstrate that the upregulation and cytoplasmic burst of the clock negative limb gene transcripts into the cytoplasm following rewarming forces a phase and amplitude change in the circadian rhythm (Fig 5). This is a critical finding as it demonstrates the observed transcriptome changes are of physiological importance and uncovers a novel mechanism by which temperature influences the cellular circadian clock. There is an intricate relationship between the circadian clock and temperature. For example, repeated cycling between high and low temperatures within the narrow physiological temperature range typically experienced by homeothermic mammals (33–37°C) is sufficient to entrain circadian rhythms (Brown et al, 2002; Saini et al, 2012). RBM3 and CIRBP, activated by such mild temperature changes, are key regulators of this process (Morf et al, 2012; Liu et al, 2013). Similarly, exposing cells to a heat shock causes circadian rhythms to reset in a HSF1-dependent manner (Reinke et al, 2008; Buhr et al, 2010; Tamaru et al, 2011). However, neither of these pathways are likely to account for the observed phase changes associated with exposure to 18°C and subsequent rewarming as neither CIRBP, RBM3, nor heat-shock response gene transcript levels are upregulated (Appendix Table S1). Instead, we identified a novel mechanism by which a phase change is triggered by a cold-induced impairment of nuclear export of upregulated clock transcripts and their sudden release into the cytoplasm after rewarming. How the expression of the clock negative limb genes is upregulated in the nucleus is unclear but may be dependent on a temperature-specific influx of calcium that could activate calmodulin–CaMKII kinase and so boost E-box gene expression by promoting

dimerization of CLOCK and ARNTL (Kon *et al*, 2014). Our data further show that the cold-induced upregulation of REV-ERBα is required to trigger a phase shift after a 5 h 18°C exposure. This is of high significance as it not only identifies REV-ERBα as an important cold-induced gene but it also demonstrates that massive upregulation of REV-ERBα can control phase resetting in mammalian cells.

How the cold-induced upregulation of REV-ERBα forces resetting of the clock is unclear. We have no evidence that the accumulation of *REV-ERBα* acts via the stabilizing loop as both *ARNTL* and *CLOCK* RNA levels, at least at the time points sampled, do not drop significantly (Fig EV3F and G). It is thus more likely that the effects of the cold-induced accumulation of REV-ERBα are more complex and include disruption of alternative pathways. Interestingly, REV-ERBα has recently been shown to be one of many nuclear receptors that interact with PER2 forming assorted complexes that differentially modulate ARNTL activity (Schmutz *et al*, 2010). Perhaps the sudden accumulation of REV-ERBα after 18°C exposure imbalances the composition of these complexes, which in turn may affect the expression of circadian genes. Furthermore, it is plausible that the sudden increase of REV-ERBα may similarly influence other PER2 controlled mechanisms such as the localization (Lee *et al*, 2001) and stability (Xing *et al*, 2013; Yoo *et al*, 2013) of CRY proteins.

Finally, given that REV-ERBα is a key chromatin regulator (Kim *et al*, 2018) linking circadian rhythms with metabolism (Cho *et al*, 2012; Zhang *et al*, 2015), inflammation (Pariollaud *et al*, 2018), and cardiac surgery outcomes (Montaigne *et al*, 2018) and has tumor suppressor (Sulli *et al*, 2018) and anti-viral properties (Zhuang *et al*, 2019), the recognition that *REV-ERBα* expression can be boosted more than 10-fold by 18°C exposure will be of significance for the development of novel strategies that broaden the applications of controlled deep hypothermia.

# Material and Methods

## Cell culture

AC16 (Davidson *et al*, 2005) cells were cultured in Dulbecco's Modified Eagle's Medium: Nutrient Mixture F-12 (DMEM/F-12), containing 10% (*v/v*) Fetal Bovine Serum (FBS), 2.5 mM ʟ-Glutamine and 1% Penicillin/Streptomycin (Sigma cat#P0781).

We used a U2OS osteosarcoma cell line stably expressing the luciferase reporter gene (*luc2* (*Photinus pyralis*)) under the control of the mouse *PER2* promoter (U2OS *PER2::LUC*) (Jagannath *et al*, 2013). U2OS cells were cultured in DMEM containing 10% (*v/v*) FBS, 2.5 mM L-Glutamine, and 1% Penicillin/Streptomycin (Sigma cat#P0781) except during bioluminescence recording (see below).

Unless otherwise specified, human cells were cultured in a 37°C incubator with 5% $CO_2$ to 80–90% confluency before harvesting, passaging or transfer to incubators set at 8, 18°C, or 28°C (also with 5% $CO_2$).

*Drosophila melanogaster* Schneider 2 (S2) cells were cultured in Schneider's Drosophila Medium (Thermo Fisher, cat.no. 21720024), 10% heat-inactivated FCS, and 1% Penicillin/Streptomycin (Sigma cat#P0781) in a 28°C incubator with 5% $CO_2$ to 80–90% confluency before harvesting. Cells were aliquoted, pelleted, and flash frozen. Aliquots harvested at the same time were used in all spike-in experiments.

## Viability assay

To assess the viability of AC16 cells after exposure to 28°C, 18°C, or 8°C, flasks of cells were exposed to each temperature for 24 h or to each temperature for 24 h and then returned to 37°C for 24 h. Adherent cells remaining after exposure to each condition were harvested and counted, and a ratio of the number cells relative to the number in a flask prior to cooling was calculated (Appendix Fig S3).

## Genome editing

### *REV-ERBα* deletion

sgRNAs were cloned into the pSpCas9(BB)–2A-Puro(PX459)-V2.0 vector (Addgene #62988) by annealing oligonucleotides 1 and 2 (Appendix Table S3) and inserting them into the BbsI site, as previously described (Ann Ran *et al*, 2013). For *REV-ERBα* knockout, two sgRNAs, sgRNA Del US and sgRNA Del DS, were designed to cleave 723 bp upstream and 135 bp downstream of the start codon, respectively, to create an approximate 878 bp deletion of a region containing the promoter and first exon of *REV-ERBα*. This leads to a frame shift of subsequent exons, which is likely to disrupt REV-ERBα function. Cloned vectors (1.5 μg each per 6-well plate) were transfected using Lipofectamine 3000 (Thermo Fisher) according to the manufacturer's instructions into U2OS *PER2::LUC* cells. As this cell line is already puromycin resistant, no selection for transfected cells was possible. Individual colonies were screened by PCR to identify correctly targeted homozygous clones.

### *REV-ERBα*-3xFLAG tagging

For *REV-ERBα* 3xFLAG C-terminal tagging, one sgRNA, sgRNA Tag, was designed to cleave 3 bp downstream of the stop codon. The cloned vector (3.5 μg) was transfected into AC16 cells with 1.5 μl single-stranded oligonucleotide (ssODN) (10 μM) as a template for homology-directed repair (HDR)-mediated 3xFLAG tag knock-in (Appendix Table S3). The ssODN was designed with asymmetric homology arms, which have been proposed to enhance HDR efficiency (Richardson *et al*, 2016). Transfected cells were selected by incubation with puromycin (1 μg/ml) in DMEM for 48 h, and individual colonies were screened by PCR and Western blot to identify correctly targeted homozygous clones.

## Nuclear and cytoplasmic RNA extraction

Extraction of RNA from nuclear and cytoplasmic subcellular fractions was carried out as described in (Neve *et al*, 2016). $1 \times 10^7$ AC16 cells were trypsinized and washed twice in ice cold PBS. Cytoplasmic membranes were lysed by slowly resuspending the cell pellet in 1 ml Lysis Buffer A (10 mM Tris–HCl (pH 8–8.4), 0.14 M NaCl, 1.5 mM $MgCl_2$, 0.5% NP-40). Nuclei were pelleted (1,000 *g*, 4°C, 3 min) and the supernatant, consisting of the cytoplasmic fraction, was cleared of remaining nuclei by spinning (11,000 *g*, 4°C, 1 min). The pelleted nuclei were resuspended in 1 ml Lysis Buffer A. 100 μl Detergent Stock Solution (3.3% (*w/v*) sodium deoxycholate, 6.6% (*v/v*) Tween 40) was then added dropwise under slow vortexing. Stripped nuclei were pelleted (1,000 *g*, 4°C, 3 min), washed once with 1 ml Lysis Buffer A, and then resuspended in 1 ml TRIzol (Thermo Fisher). 500 μl of TRIzol was also added to

500 µl of the cleared cytoplasmic fraction. RNA was purified from the nuclear and cytoplasmic fractions as per the TRIzol manufacturer's instructions. RNA samples were DNase I-treated (Roche) as per the manufacturer's instructions, purified using phenol-chloroform, and precipitated with ethanol.

For *D. melanogaster* Schneider 2 (S2) cell spike-in experiments, S2 and AC16 cells were mixed at a known ratio (either 1:4 (replicates 1 and 2) or 5:4 (replicates 3 and 4)) before cell lysis.

## RNA-Seq

The number of replicates sequenced for each condition is listed in Appendix Table S4.

### QuantSeq

Barcoded libraries for multiplexed, strand-specific sequencing of the 3′ end of polyadenylated RNAs were generated using the QuantSeq 3′ mRNA-Seq Library Prep Kit for Ion Torrent (Lexogen) as per the manufacturer's instructions, using 500 ng and 1,700 ng input RNA for nuclear and cytoplasmic RNA samples, respectively, and using 13 PCR cycles. Libraries were loaded onto the Ion Chef System (Thermo Fisher) for template preparation, and chip loading and the resulting chips were sequenced on the Ion Proton Sequencing System (Thermo Fisher) as per the manufacturer's instructions.

For standard libraries, reads were aligned to the hg19 genome build using the Ion Torrent Server TMAP aligner with default alignment settings (-tmap mapall stage1 map4). Human polyA site (PAS) annotations were obtained from PolyA_DB 3 (Wang *et al*, 2018). Each PAS was extended 20 nt 3′ and 200 nt 5′ from the site of cleavage and those that overlapped on the same strand after extension were combined into a single PAS annotation. Mapped reads were narrowed to their 3′ most nucleotide and those which overlapped with the extended PAS annotations were counted. Counts were then obtained for each gene by combining the counts for all PASs associated with each gene. Genes not in the RefSeq gene database were excluded.

For libraries prepared from *D. melanogaster* Schneider 2 (S2) cell spiked in samples, reads were simultaneously aligned to both human and *D. melanogaster* genomes by aligning to a custom combined hg19 and dm6 genome build using the TMAP aligner. *D. melanogaster* PASs were obtained from the Tian laboratory (Liu *et al*, 2017) and were extended, in the same way as for human PASs. Mapped reads were narrowed to their 3′ most nucleotide and those which overlapped with the extended human or *D. melanogaster* PAS annotations were counted. The ratio of total *D. melanogaster* PAS to human PAS-associated reads was obtained for each sample. These were then scaled relative to each other according to the ratio at which the two cell lines were combined.

### Full-length RNA-Seq

Barcoded libraries for multiplexed, full-length, strand-specific sequencing of polyadenylated RNAs were generated using the Ion Total RNA-Seq Kit v2 (Thermo Fisher) as per the manufacturer's instructions. Polyadenylated RNAs were first purified from total RNA from each fraction using the NEBNext Poly(A) mRNA Isolation Module as per the manufacturer's instructions. Libraries were sequenced and aligned in the same way as the QuantSeq libraries. Data were normalized to sequencing depth by calculating the count

at each nucleotide per million counts aligned at all nucleotides (CPM). CPM were converted to bigwig files and visualized using Integrative Genomics Viewer (IGV) (Thorvaldsdóttir *et al*, 2013).

## QuantSeq data processing

### Differential expression analysis

Differential expression for each pairwise comparison was assessed using the DESeq algorithm within the DESeq2 R package (Love *et al*, 2014). For each gene, this gives a base mean value, which is the mean of normalized counts for all samples after normalizing for sequencing depth, a value, and standard error for the $\log_2$ fold change in relative RNA levels between the two samples being compared and a *P*-value determined using a Wald test, adjusted for multiple testing using the Benjamini–Hochberg method (padj), showing the significance of any difference.

### Half-life comparison

Published genome-wide mRNA half-life data, determined using a metabolic labeling approach with the nucleoside analog 4-thiouridine (4SU) in HEK293T cells, were used (Lugowski *et al*, 2018). The $\log_2$ fold change in cytoplasmic RNA level for each gene was taken from the DESeq algorithm output for the comparison between cells kept at 37°C and cells transferred to 28, 18, or 8°C for 24 h. In order to reduce noise and identify a relationship with mRNA half-life, data were processed as follows: Genes with missing half-life or $\log_2$ fold change values in any of the three temperature comparisons were excluded. Genes with extreme half-lives were excluded by removing those with a $\log_2$ half-life greater than two standard deviations away from the mean $\log_2$ half-life. For each cold temperature, genes were sorted by the $\log_2$ fold change for that temperature and split into 10 equal-sized groups. The mean $\log_2$ fold change and mean $\log_2$ half-life were then calculated for each group and plotted against each other. A linear regression line, Pearson correlation coefficient and *P*-value testing the significance of the correlation were calculated for each comparison.

### Density scatter graphs comparing RNA level changes at different cold temperatures

Density scatter graphs were plotted showing the relationship in both the nucleus and cytoplasm between changes in RNA level upon transfer of cells from 37 to 28°C or 18, 28°C or 8°C, and 18°C or 8°C for 24 h. This is done using the $\log_2$ fold change in RNA level for each gene from the DESeq algorithm output for the comparison between cells transferred to a specific cold temperature and cells kept at 37°C. Positive values show genes with higher relative RNA levels at the cold temperature. Genes with no padj or $\log_2$ fold change value in either of the two cold temperature vs. 37°C comparisons were excluded.

### Density scatter graphs comparing nuclear and cytoplasmic RNA level changes

Density scatter graphs were plotted showing the relationship between changes in nuclear RNA levels and changes in cytoplasmic RNA levels from cells kept at 37°C and cells transferred from 37°C to a specific cold temperature for 5, 10, or 24 h. This is done using the $\log_2$ fold change in RNA level for each gene from the DESeq algorithm output for the comparison between nuclear fractions and

for the comparison between cytoplasmic fractions. Positive values show genes with higher relative RNA levels at the cold temperature. Genes with a base mean value less than 30 in either the cytoplasmic or nuclear comparisons were excluded.

### Standardized RNA level boxplots

RNA level counts for all genes for all samples from each time series were normalized by applying a regularized log transformation using the "rlogTransformation" function from the DESeq2 package with the blind option set to FALSE. The mean normalized count for each gene was then calculated across all repeats for each time point. The counts for each gene were then standardized so that the mean and standard deviation of the RNA level for each gene across the time series were zero and one, respectively. This process makes the expression profiles of different transcripts across the time series comparable and was carried out separately for the nuclear and cytoplasmic fraction samples. Box and whisker plots displaying the range (whiskers), interquartile range (IQR, box edges), median ± IQR/[square root (number of genes)] (notch), and median (central line) of these standardized RNA levels were then plotted for nuclear and cytoplasmic fractions across each time series for subsets of genes selected according to whether they show significant ($Padj < 0.05$) up or downregulation at a particular timepoint in the nuclear fraction.

### GO term analysis

Genes showing significant ($Padj < 0.05$) upregulation in the cytoplasm after exposing cells to 18°C for 24 h followed by 37°C for 2 h were submitted for gene ontology (GO) analysis using the KEGG 2019 database within Enrichr (Kuleshov *et al*, 2016).

### Reads per million bar graphs

Read counts for each extended PAS were normalized for sequencing depth by dividing by the total number of PAS associated reads for each sample and then multiplying by 1 million to give reads per million (RPM). RPM were then obtained for each gene by combining the RPM for all PASs associated with each gene. RPM were then averaged across all replicates for each condition.

## Western blot

$1 \times 10^6$ cells were trypsinized and washed with ice cold PBS. To assess whole cell protein levels, cell pellets were resuspended in 100 μl Lysis Buffer A, incubated (37°C, 20 min) with 0.5 μl Benzonase (Merck), and boiled (5 min) with 33 μl 4× Laemmli Buffer.

Protein samples were separated by gel electrophoresis on 10% SDS polyacrylamide gels and transferred to nitrocellulose membranes using semi-dry transfer apparatus. Membranes were Ponceau stained, imaged, and then washed in TBST (20 mM Tris–HCl (pH 7.5), 150 mM NaCl, 0.1% Tween-20). Membranes were then incubated (4°C, overnight) with 5% milk powder (Bio-Rad) in TBST, then primary antibody in 5% milk powder in TBST for 1.5 h, washed, then incubated with rabbit or mouse (Promega) HRP-conjugated secondary antibody diluted 1:4,000 in 5% milk powder in TBST for 45 min and washed again. Primary antibodies and their dilutions are detailed in Appendix Table S5. Antibody binding was visualized using chemiluminescence (Pierce) and X-ray film.

## RNA fluorescence *in situ* hybridisation (RNA-FISH)

### RNA-FISH slide preparation

Cells were cultured on poly-L-lysine-treated coverslips. Culture media was aspirated, and coverslips were washed once with PBS. Cells were fixed by incubating for 10 min with 4% formaldehyde/PBS, washed twice with PBS, and permeabilized by incubating (> 3 h, −20°C) in 70% ethanol. Cells were rehydrated by incubating (5 min, RT) with FISH wash buffer (10% formamide, 2× SSC). For hybridization, coverslips were placed cell-coated side down on a 48 μl drop containing 100 nM Quasar 570-labeled probes complementary to one of *REV-ERBα*, *CRY2*, or *TP53* transcripts (Biosearch Technologies) (see Appendix Table S6 for probe sequences), 0.1 g/ml dextran sulfate, 1 mg/ml *Escherichia coli* tRNA, 2 mM VRC, 20 μg/ml BSA, 2× SSC, and 10% formamide and incubated (37°C, 20 h) in a sealed parafilm chamber. Coverslips were twice incubated (37°C, 30 min) in prewarmed FISH wash buffer, then in PBS containing 0.5 μg/ml 4′,6-diamidino-2-phenylindole (DAPI) (5 min, RT), washed twice with PBS, dipped in water, air-dried, placed cell-coated side down on a drop of ProLong Diamond Antifade Mountant (Life Technologies), allowed to polymerize for 24 h in the dark, and then sealed with nail varnish.

### RNA-FISH image acquisition and analysis

Cells were imaged using a DeltaVision Elite wide-field fluorescence deconvolution microscope using a 60×/1.40 objective lens (Olympus) and immersion oil with refractive index 1.514. First, the Quasar570-labeled RNA-FISH probes were imaged using the TRITC channel, followed by imaging of the DAPI channel. The number of 0.2 μm-separated stacks imaged, number of replicates imaged for each condition, and number of images across all combined replicates are listed in Appendix Table S7 (column "Stacks", "Reps", and "Total images", respectively). Images were deconvolved using the default conservative deconvolution method using DeltaVision soft-WoRx software. Image quantification was carried out using Fiji (Schindelin *et al*, 2012). Deconvolved images were compressed to 2D images displaying the maximum intensity projection for each pixel across *z*-stacks listed in Appendix Table S7 (column "Projected"). Cell and nuclear areas were outlined using thresholding functions on the background TRITC signal and DAPI signal, respectively. Dots corresponding to transcripts were then counted for both nuclear and cytoplasmic areas for each image by applying the "Find Maxima" command with a noise tolerance specified in Appendix Table S7 (column "Maxima"). Bar charts show the mean number of dots per nuclear area and cytoplasmic area across all images for all combined replicates.

## Luciferase reporter assay

Two repeats of the six distinct profile experiments for both the 18°C for 5- and 24-h incubations and 1 repeat of the four distinct profile experiment for an 18°C for 5-h incubation were carried out.

### Six distinct profile experiment

In triplicate (i.e., 3 × 96 well plates), U2OS *PER2::LUC* normal (WT cells) and U2OS *PER2::LUC REV-ERBα* KO cells (KO cells) were each seeded at 20% confluency in 30 wells of 96-well plates. After incubation for 24 h at 37°C, cells within a well were

synchronized by incubating (45 min, 37°C) with 100 μl recording media (per liter: DMEM powder (Sigma cat#D2902), 1.9 g glucose, 10 mM HEPES, 1× GlutaMAX (Thermo Fisher), 2.5 ml Penicillin/Streptomycin (Sigma cat#P0781), 4.7 ml NaHCO$_3$ (7.5% solution), and final pH 7.3), containing 100 nM dexamethasone. Ten wells of each cell type in each plate were treated at one of three time intervals separated by 4 h to give three groups of wells staggered at three different phases of the circadian cycle. After incubation, cells were washed twice with 100 μl prewarmed PBS and then 100 μl prewarmed recording media supplemented with 1× B-27 (Thermo Fisher) and 100 μg/ml Luciferin was added. After all wells had been treated, all plates were sealed and incubated at 37°C in plate readers, while recording the bioluminescence at 30-min intervals. One control plate was kept at 37°C for at least 144 h and used as the control for both test plates. Test plates one and two were transferred to 18°C after 60 h and 72 h, respectively, after the start of the first dexamethasone treatment. This ensures that there are six groups of 10 wells of cells at six distinct phases of the circadian cycle at the point of 18°C exposure. Both test plates were returned to 37°C plate readers after either 5 h or 24 h, and the bioluminescence was again recorded at 30-min intervals. Two repeats of this experiment were carried out for both 5- and 24-h incubations. No control plate was recorded for the second repeat of the 5-h incubation.

### Four distinct profile experiment

The same as for the six distinct profile experiment, except 15 wells of each cell type in each plate were treated at one of two time intervals separated by 12 h to give two groups of wells staggered at two different phases of the circadian cycle. Test plates one and two were transferred to 18°C after 60 and 66 h, respectively, after the start of the first dexamethasone treatment. This ensures that there are four groups of 15 wells of cells at four distinct phases of the circadian cycle at the point of 18°C exposure. Both test plates were returned to 37°C plate readers after 5 h. One repeat of this experiment was carried out.

### Luciferase reporter assay data processing

MultiCycle (Actimetrics) gives the period length and amplitude of recordings, and these values were taken for all control wells across all control plates (plates kept at 37°C). The fold change in amplitude between WT and KO cells was calculated by dividing the amplitude of each well containing KO cells by that of the corresponding well in the same row containing WT cells within the same plate.

Two out of the total 540 individual well bioluminescence recordings across all experiments were discarded for technical errors during experimental setup. Raw bioluminescence recordings from all plates were baseline-detrended using MultiCycle. These were used to calculate the mean and standard deviation profiles for each group of synchronized plate wells. For all the six distinct profile experiments, all WT cell profiles were scaled to the same WT cell control mean profile so that their profile amplitudes for the period before the cold exposure were the same. All KO cell profiles were similarly scaled to the same KO cell control mean profile so that their profile amplitudes for the period before the cold exposure were also the same. The same scaling procedure was applied to the profiles from the four distinct profile experiment.

For scatter plots showing the change in phase variance resulting from 18°C incubations, for each well profile, the time to the second peak in bioluminescence after the start of the cold exposure and the equivalent peak in control well plates (this is the second peak after the end of the cold exposure for 24-h incubations) was determined. These times were then plotted for each condition and colored according to their distinct phase prior to 18°C exposure. This shows the effect of 18°C exposure on the spread of these profiles in comparison with profiles from cells kept at 37°C for both WT and KO cells. $P$-values (adjusted for multiple testing using the Holm–Bonferroni method) were calculated using the Brown–Forsythe method to test the significance of any difference in the variance for pairwise comparisons between 18°C-exposed cells and cells kept at 37°C and between WT and KO cells.

The same peak and the subsequent trough were used to calculate amplitude fold changes and phase shifts for profiles from cold-exposed wells relative to the corresponding mean control profile. Phase shifts as a percentage of period length were calculated using the following formula: [(peak time$_{cold\text{-}exposed}$ + trough time$_{cold\text{-}exposed}$) − (peak time$_{control}$ + trough time$_{control}$)]/(2 × mean period length). Mean shifts were calculated for all well profiles from each group with a distinct phase prior to 18°C exposure and for each experiment replicate. A one-sided, one sample t-test was then used to determine mean shifts significantly ($P$-value (adjusted for multiple testing using the Holm–Bonferroni method) < 0.05) greater than 5% in either direction. Amplitude fold changes were calculated using the following formula (peak bioluminescence signal$_{cold\text{-}exposed}$ − trough bioluminescence signal$_{cold\text{-}exposed}$)/(peak bioluminescence signal$_{control}$ − trough bioluminescence signal$_{control}$). Mean log$_2$ transformed fold changes were calculated for all well profiles from each group with a distinct phase prior to 18°C exposure and for each experiment replicate. A one-sided, one sample t-test was then used to determine mean changes significantly ($P$-value (adjusted for multiple testing using the Holm–Bonferroni method) < 0.05) greater or less than + or −, respectively, log$_2$(1.4).

For experiments involving an 18°C for 5-h incubation, the position of the control profile at the point of rewarming within the circadian period of PER2::LUC expression was determined. As the phase of the REV-ERBα expression profile is shifted approximately 6 h in relation to PER2 expression in U2OS cells (Hoffmann et al, 2014), it is possible to predict the approximate level and direction of change of REV-ERBα expression at this point. Each mean phase shift and log$_2$ amplitude fold change were plotted to approximately align with the position of its control profile at the point of rewarming within a circadian period represented by sine curves for PER2:LUC expression and predicted REV-ERBα expression.

### 3D-structured illumination microscopy (3D-SIM)

#### 3D-SIM slide preparation

AC16 cells were cultured on 22 × 22 mm #1.5H high precision 170 μm ± 5 μm poly-L-lysine-treated coverslips (Marienfeld Superior). After exposure to a particular temperature condition, culture media was aspirated, and coverslips were washed once with PBS. Cells were fixed by incubating for 10 min in 2% formaldehyde/PBS, then washed with 0.05% Tween-20/PBS (PBST), permeabilized by incubating for 10 min in 0.2% Triton X-100/PBS, and then washed again with PBST. Coverslips were incubated (30 min, RT) in

MaxBlock (ActiveMotif, cat#15252), then (overnight, 4°C) in MaxBlock containing primary antibodies against the proteins of interest (primary antibody details and dilutions are shown in Appendix Table S8), washed 4 times with PBST, then incubated (30 min, RT) in MaxBlock containing fluorescently labeled secondary antibodies (secondary antibody details and dilutions are shown in Appendix Table S8), and washed again four times with PBST. Cells were post-fixed by incubating for 10 min in 4% formaldehyde/PBS and washed with PBST. Coverslips were incubated in PBST containing 2 μg/ml DAPI, washed with PBST, mounted in non-hardening Vectashield (Vector Laboratories, cat#H-1000), and stored at 4°C.

### 3D-SIM image acquisition, reconstruction, and quality control

3D-SIM image acquisition, reconstruction, and quality control were carried out as detailed in preprint: Miron *et al* (2019). Images were acquired with a DeltaVision OMX V3 Blaze system (GE Healthcare) equipped with a 60×/1.42 NA PlanApo oil immersion objective (Olympus), pco.edge 5.5 sCMOS cameras (PCO) and 405, 488, 593, and 640 nm lasers. Spherical aberration was minimized using immersion oil with refractive index (RI) 1.514. 3D image stacks were acquired over the whole nuclear volume in *z* and with 15 raw images per plane (five phases, three angles). The raw data were computationally reconstructed with SoftWoRx 6.5.2 (GE Healthcare) using channel-specific OTFs recorded using immersion oil with RI 1.512, and Wiener filter settings between 0.002 and 0.006 to generate 3D stacks of 115 nm (488 nm) or 130 nm (593 nm) lateral and approximately 350-nm axial resolution. Multi-channel acquisitions were aligned in 3D using Chromagnon software (Matsuda *et al*, 2018) based on 3D-SIM acquisitions of multi-color EdU-labeled C127 cells (Kraus *et al*, 2017). Appendix Table S9 details the number of nuclei imaged for each repeat, temperature condition, and marker combination.

3D-SIM imaging and subsequent quantitative analyses are susceptible to artefacts (Demmerle *et al*, 2017). For instance, bulk labeling of densely packed chromatin inside mammalian nuclei of several μm depth entails high levels of out-of-focus blur, which reduces the contrast of illumination stripe modulation and thus the ability to recover high-frequency (i.e., super-resolution) structure information. All SIM data were therefore routinely and meticulously quality controlled for effective resolution and absence of artifacts using SIMcheck (Ball *et al*, 2015), an open-source ImageJ plugin to assess SIM image quality via modulation contrast-to-noise ratio (MCNR), spherical aberration mismatch, reconstructed Fourier plot, and reconstructed intensity histogram values (for more details see Demmerle *et al*, 2017).

To exclude potential false-positive calls, we used the "MCNR map" macro of SIMcheck, which generates a metric of local stripe contrast in different regions of the raw data and directly correlates with the level of high-frequency information content in the reconstructed data (Demmerle *et al*, 2017). Only IF centroid signals with the underlying MCNR values exceeding a stringent quality threshold were considered, while localisations with low underlying MCNR were discarded during the ChaiN analysis pipeline to exclude any SIM signal which falls below reconstruction confidence.

### ChaiN—pipeline for high-content analysis of the 3D epigenome

ChaiN (for Chain high-throughput analysis of the *in situ* Nucleome), as detailed in preprint: Miron *et al* (2019), consists of a pipeline of scripts for the automated high-throughput analyses presented in this investigation. Scripts are available from https://github.com/eze miron/Chain. In brief, the DAPI-stain/chromatin channel is used to generate a nuclear mask and segment chromatin topology into seven intensity/density classes using an R script that is expanding on a Hidden Markov model (Schmid *et al*, 2017). Class 1 denotes the inter-chromatin region while classes from 2 and 7 denote chromatin with increasing intensity/density. Based on this output of ChaiN, we quantitatively assessed the overall volume of chromatin (combined classes 2–7) vs. the total nuclear volume (combined classes 1–7). Boxplots for nuclear volume and chromatin:nuclear volume display the median (central line), interquartile range (IQR, box edges), and most extreme data point no more than 1.5× IQR from box edges (whiskers). Outliers outside the whiskers are shown as individual points.

The IF signals for markers are thresholded by intensity using the Otsu algorithm, with all non-foci voxels replaced by zeros. The complement of this image is segmented by a 3D watershed algorithm and the 3D centroid coordinates extracted from each segmented region's center of mass after weighting by local pixel intensities. Potential false-positive localizations were then filtered using the MCNR threshold detailed above to avoid potential artefacts skewing the statistical output. The quality controlled and filtered centroid positions were then related to the chromatin intensity/density classes. Centroid positions falling into each of the seven segmented classes were counted, normalized to the respective class size and their relative enrichment (or depletion) displayed in a heat map on a $\log_2$-fold scale.

Nearest neighbor analysis was performed as follows. For each nucleus imaged, the Euclidian distance between each NPC centroid and all other hnRNPC centroids was calculated. The shortest of these distances was then taken and compiled into a list containing a distance for each NPC from all nuclei and from all temperature conditions. To remove outliers, distances greater than 2 standard deviations from the mean were excluded. Mean and SEM distance was then calculated for each temperature condition.

## Data availability

RNA-Seq data: Gene Expression Omnibus, accession code: GSE137003 (https://www.ncbi.nlm.nih.gov/geo/query/acc.cgi? acc = GSE137003).

Deconvolved RNA-FISH images and reconstructed 3D-SIM images: Image Data Resource, accession number: id0089 (idr.open microscopy.org).

Codes for RNA-FISH and transcriptomic analyses: Github (github.com/hjf34/Cold).

Expanded View for this article is available online.

## Acknowledgements

We thank Paul Riley, Robin Choudhury, all members of the Furger and Mellor laboratories for helpful discussions and Nick Proudfoot, Shona Murphy, and Andy Baldwin for comments on the manuscript. This research is funded by grants from the BBSRC awarded to AF (BB/N001184/1) and JM and AF (BB/S009035/1) and AJ (BB/N01992X/1 BBSRC, David Phillips fellowship). Imaging was performed at the Micron Oxford Advanced Bioimaging Unit funded by a Wellcome Trust Strategic Award (091911 and 107457/Z/15/Z).

## Author contributions

AF conceptualized and administered the project and supervised research planning. JM supervised RNA-FISH analysis. LS supervised 3D-SIM imaging experiments. AJ supervised the circadian rhythm experiments. HF was responsible for formal analysis, design, and experimentation and wrote all in house-developed scripts (except 3D-SIM-related algorithms). RO and LS supported HF in 3D-SIM. DM supported HF in isolation and sequencing of RNA and analysis of RNA-Seq data. AF and HF wrote the manuscript with contributions from all other authors, and HF created visualizations of the data with inputs from AF, JM, LS, and AJ.

## Conflict of interest

The authors declare that they have no conflict of interest.

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
