## [Review Process File · The EMBO Journal]

Cold-induced chromatin compaction and nuclear retention of clock mRNAs resets the circadian rhythm

Harry Fischl, David McManus, Roel Oldenkamp, Lothar Schermelleh, Jane Mellor, Aarti Jagannath, and Andre Furger

DOI: [10.15252/embj.2020105604](https://doi.org/10.15252/embj.2020105604)

Corresponding author(s): Andre Furger (andre.furger@bioch.ox.ac.uk)

Review Timeline:

Submission Date:	12th May 20
Editorial Decision:	10th Jun 20
Revision Received:	3rd Aug 20
Editorial Decision:	19th Aug 20
Revision Received:	28th Aug 20
Accepted:	29th Aug 20

Editor: Stefanie Boehm

Transaction Report:

Dr. Andre Furger
University of Oxford
Biochemistry
South Parks Road
Oxford OX1 3RE
United Kingdom

Stefanie Boehm
Editor
The EMBO Journal

10th Jun 2020

Re: EMBOJ-2020-105604

Cold induced chromatin compaction and nuclear retention of clock mRNAs resets the circadian rhythm

Dear Dr. Furger,

Thank you for submitting your manuscript for consideration by The EMBO Journal. Please also excuse the delay in communicating this decision to you, which was due to a delayed review process on account of the current pandemic. We have now however received three referee reports on your study, which are included below for your information.

As you will see, the reviewers are overall positive and express an interest in the study. Nonetheless they also raise some concerns that would need to be addressed in a revised manuscript. In particular, referee #2 (point 1) and referee #3 (point 3) find that additional discussion of the observed effects on chromatin and the nuclear transcriptome at 8{degree sign}C is needed. In addition to discussing this further in the revised version, please also address the comments of referee #1 regarding REV-ERB α in context of the current models for circadian rhythms. Moreover, referee #2 also pointed out several instances where statistical analyses or information regarding the number of repeats is missing (point 2, 4, 8), or where additional data points are required to support the conclusion drawn (point 6, 9). Please include the statistical information and add the respective date to Fig. 3 and Fig. 4 and/or revise the text. Furthermore, please also discuss the potential cell line specificity (ref#2- 10) and clarify if the effects at later time points can be observed across different cell lines (ref#2- 11), as well as carefully considering and responding to all other comments the referees raise.

Please note that it is our policy to allow only a single round of major revision. We realize that lab work worldwide is currently affected by the COVID-19/SARS-CoV-2 pandemic and that an experimental revision may be delayed. We can extend the revision time when needed, and we have extended our 'scooping protection policy' to cover the period required for a full revision. However, it is nonetheless important to clarify any questions and concerns at this stage and we encourage you to discuss a revision plan and any potential issues you may foresee as soon as possible.

Please also feel free to contact me should you have any other further questions. Thank you for the opportunity to consider your work for publication. I look forward to receiving your revised manuscript.

Kind regards,

Stefanie Boehm

Stefanie Boehm
Editor
The EMBO Journal

Referee #1:

The effect of temperature on circadian clocks is highly complex. While these timekeepers are temperature-compensated even in homeothermic animals, they can be synchronized even by minute oscillations in temperature within the physiological temperature range (Saini et al., 2012). These temperature oscillations are accompanied by the rhythmic shuttling of Heat Shock Factor (HSF1) between nucleus and cytoplasm (Reinke et al., 2008) and the cyclic expression of the cold-inducible RNA-binding protein CIRP (and to a lesser extent RBM3) (Morf et al., 2012), which have been shown to participate in the temperature-dependent phase resetting.

Temperatures below the physiological range are used in the clinic during certain surgical interventions and for the preservation of organs destined for transplantation. In this interesting study the authors examined the effect of sub-physiological temperatures on chromatin compaction and circadian clock gene expression in two human cell lines (AC16, U2OS). Surprisingly, they find that chromatin assumes a more condensed state when lowering the temperature from 37{degree sign}C to 18{degree sign}C (sometimes used during controlled deep hypothermia in the clinic), but shows little changes at 8{degree sign}C (used for the storage of organs) when compared to 37{degree sign}C. These changes are reversible, as chromatin topology rapidly assumes the normal state upon resetting the temperature to 37{degree sign}C. The temperature-dependent transitions in chromatin structure do not appear to involve posttranslational modifications of histones or RNA polymerase II. Using cell fractionation and FISH experiments the authors obtained compelling evidence that cooling cells to 18{degree sign}C engenders a retention of polyadenylated transcripts in the nucleus, probably owing to different configurations of perinuclear chromatin and nuclear pores. Again, this nuclear transcript retention is reversible, in that it is rapidly alleviated upon readjusting the temperature to 37{degree sign}C. The authors then went on to study the impact of transcript release into the cytoplasm for circadian gene expression upon rewarming the cells from 18{degree sign}C to 37{degree sign}C. Surprisingly, they observed that mRNAs encoding core clock proteins enter the cytoplasm with different kinetics during this process. Among the examined transcripts, REV-ERB α mRNA is exported most rapidly into the cytosol, and this elicits a surge of REV-ERB α protein accumulation. The authors then compared the phases of circadian PER2::luc expression in U2OS cells with or without functional REV-ERB α alleles in a clever set of synchronization experiments (Figure 5). Briefly, PER2::luc expression was synchronized to different phases by dexamethasone in parallel cultures. These were then exposed to the cooling-rewarming protocol, and the phase-resetting was examined. In experiments in which the cooling to 18{degree sign}C was restricted to 5 hours (i.e. a time period during which newly synthesized REV-ERB α transcripts remained largely in the nucleus), only wild-type cells could efficiently set all differently pre-synchronized phases to the same phase. Hence, under these conditions the surge of REV-ERB α expression upon re-warming was required to set all phases to a nearly identical one.

General Comments

This is an interesting study on the effects of sub-physiological temperature on circadian gene expression that have hitherto not yet been investigated. A question that remains open is how a surge of REV-ERB α accumulation can synchronize clock gene expression without affecting CLOCK

and BMAL1 expression. REV-ERB α has been thought to modulate the expression of positive limb members (i.e.), however, the authors did not observe temperature-dependent changes in CLOCK and BMAL1 mRNAs. It is thus unclear how the release of REV-ERB α mRNA from the nucleus can quickly synchronize the different phases established by Dexamethason (see below). In the Introduction and Discussion sections the description of the Transcriptional/Translational Feedback Loops (TTFLs) underlying circadian rhythm generation is very confusing. As it's written it gives the impression that CRYs, PERs and REV-ERBs belong to the same functional class of "repressor/corepressor" genes encoding proteins of the negative limbs of the TTFLs. According to current belief there are two coupled feedback loops driving the molecular oscillator. In the primary TTFL CRYs and PERs are assembled in large co-repressosome complexes that annul the activity of CLOCK-BMAL1 complexes. In the secondary TTFL, REV-ERB repressors cyclically inhibit the transactivation of the BMAL1 and CLOCK genes through competitive binding with ROR activators to ROREs. Since the rhythmic expression of REV-ERBs is under control of the primary TTFL, the primary and secondary TTFLs are tightly coupled. As mentioned above, the results obtained by the authors with regard to the phase-setting role of REV-ERB α in the cooling-rewarming protocol is difficult to interpret on the basis of the "coupled TTFL model". Nonetheless, the data presented in Figure 5B are rather compelling, and this reviewer thus considers likely that REV-ERB α acts through a less conventional mechanism, perhaps by binding directly to PER2 (Schmutz et al., 2010). I am not asking the authors to figure out the detailed mechanism through which Rev-ERBa synchronizes circadian clocks upon rewarming, as this would take years. However, this issue should definitely be discussed in greater detail.

Minor Issues

I have never seen the term "clock suppressor" standing for members of the negative limbs of the TTFLs in the scientific literature. While REV-ERBs act as repressors (in the classical sense) by directly binding to cis-acting regulatory elements (ROREs), CRYs and PERs can be considered as corepressors, since they bind to DNA-binding proteins (here CLOCK-BMAL1) rather than to DNA elements. It would be advisable to replace "clock suppressors" with "clock repressors/corepressors" or "negative limb members".

On page 8 (Paragraph "Cold exposure shapes...") the terms "genes" and "transcripts" are confused on multiple occasions. For example "651 and 1267 genes showing significant changes in the nucleus and the cytoplasm" should read "651 and 1267 transcripts showing significant changes in the nucleus and the cytoplasm". Please discriminate between "genes" and "transcripts" throughout the text.

On page 18 the sentence "However, both of these pathways cannot account for the observed phase changes..." should read "However, neither of these pathways can account for the observed phase changes...".

Cited References

Morf, J., Rey, G., Schneider, K., Stratmann, M., Fujita, J., Naef, F., and Schibler, U. (2012). Cold-inducible RNA-binding protein modulates circadian gene expression posttranscriptionally. *Science* 338, 379-383.

Reinke, H., Saini, C., Fleury-Olela, F., Dibner, C., Benjamin, I.J., and Schibler, U. (2008). Differential display of DNA-binding proteins reveals heat-shock factor 1 as a circadian transcription factor. *Genes & development* 22, 331-345.

Saini, C., Morf, J., Stratmann, M., Gos, P., and Schibler, U. (2012). Simulated body temperature rhythms reveal the phase-shifting behavior and plasticity of mammalian circadian oscillators. *Genes & development* 26, 567-580.

Schmutz, I., Ripperger, J.A., Baeriswyl-Aebischer, S., and Albrecht, U. (2010). The mammalian clock component PERIOD2 coordinates circadian output by interaction with nuclear receptors. *Genes & development* 24, 345-357.

Referee #2:

Paper summary

Fischl et al demonstrate that therapeutically relevant cooling temperatures leads to the upregulation of the circadian clock suppressor transcripts and that chromatin compaction limits their nuclear export. Upon rewarming, the authors show that the reversal of chromatin compaction correlates with nuclear export of transcripts. The authors propose that the release of these mRNAs to the cytoplasm leads to their translation and resets the circadian clock. These are very interesting findings. However, from Figure 2 onwards the paper was rather difficult to read as the order the data were presented was not logical order. Moreover, some of the conclusions were not supported and additional data for figures 3, 4 and 5 is required as outlined below.

Major comments

Figure 1

- Page 6 line 17/18 states "...compaction and largest increase in IC occurs at 18{degree sign}C, not 8{degree sign}C...". The chromatin landscape at 8oc looks most similar to 37oc in Fig 1A, and not like the other cooled samples. Can the authors expand on why incubating cells at 8oc does not appear to have the same effect as cooling at 28oC or 18oC?

- Page 6 line 18/19 states "...cold-induced compaction to chromatin is reversed (Fig. 1B)". However, fig 1B only shows partial reversal of compaction when incubating 18oC cells at 37oC for 2 hours. Moreover, the y-axis scale suggests that these are very moderate changes. Therefore, can the authors include statistical analysis to confirm these observations.

- What is the viability of cells grown at 8oC for 24 hours? Can the authors include analysis of cell death in each of these different cooling conditions to show that cells are not dying upon cooling and rewarming.

Figure 3

- Page 9 line 20 states "...rewarming cells back from 18{degree sign}C to 37{degree sign}C leads to a rapid readjustment of the nuclear-accumulated transcripts back to levels seen before cooling (Fig. 3A, top panel)". Can the authors include statistical analysis to support this conclusion.

- Figure 3D: It is important to show that the tagging of REV-ERBa doesn't perturb its function or mRNA localisation upon cooling and rewarming. It would also be interesting to know if the translation of REV-REBa correlates with global protein synthesis rates.

· Text on page 11 line 9/10 states "...nuclear-retained transcripts surge into the cytoplasm after rewarming to 37{degree sign}C where they are translated (Fig. 3D, S3H)". However, fig 3D and S3H only shows data for REV-ERBa and not PER1, CRY1 or CRY2. Please amend the text or include examples of other nuclear transcripts to support this text.

· Text on page 11 line 18/19 states "...we employed RNA-FISH using probes targeting transcripts for the most highly upregulated and earliest responding gene, REV-ERBa, the later responding and nuclear-restricted gene, CRY2, and the unchanged control gene, TP53 (Fig. S3F)". Fig S3F shows ARNTL, CLOCK and TP53. This is very confusing - please amend.

Figure 4

· Figure 4B and 4F - The representative images in 4B (particularly 18oC 24h, 37oC 2h) do not support the quantified data in 2F. This makes it difficult to conclude: "... after 24h, CRY2 transcripts accumulate in the nucleus and are released into the cytoplasm after rewarming". Please include the N value of replicates used to generate this quantification.

· Page 13 line 11/12 states "...rewarming following exposure to 18{degree sign}C for 5h only triggers a cytoplasmic pulse of increased REV-ERBa without affecting PER1/2 or CRY1/2 levels (Fig. 3-4)". How can the authors conclude this when the data for CRY1/2 or PER1/2 at the 5 hr time point are not included in figure 4? The authors must include the 5 hr time for at least CRY2 to make this conclusion. Moreover, figure 3B only shows the localisation of RNA for rewarming after 24 hr 18oC and not 5 hours. Please include the data for 5 hours to support this statement.

· The authors show that the localisation of REV-ERBa in the cytoplasm following 18oC 24 hr and rewarming 2 hr differs in AC16 and U2OS cells (Fig 4E and 4H - purple bar in cytoplasmic area), suggesting that REV-ERBa is retained/maintained in the cytoplasm following rewarming in U2OS cells but not AC16 cells. Could the authors provide an explanation as to why this may be the case?

Figure 5

With the differences observed across cell lines at the 24 hour time point, can the authors justify why all subsequent experiments at the 24 hr cooling time point in figure 5 were only carried out in U2OS and not in AC16 cells? The experiments at 24 hours in figure 5 should incorporate AC16 cells to ensure that the effects observed at the later time points are not cell line specific, and this would also increase the therapeutic relevance of the study.

Minor Comments

- Add N values to all figure legends.
- Figure 1B - Add missing value to axis (0.60).
- Refer to the figures in order and provide the figure reference at the appropriate place in the text.
- In figures where multiple cell lines are used, please state the cell line used in the figure legend.
- Axis titles missing in figures 3B, 3D
- Please add protein size marker label to ponceau stain in figure 3D and S3H
- Figure 3D: It is not clear if these are cytoplasmic cell extracts. Please add more detail to the figure legend.
- @ 37{degree sign}C is missing from figure S3A
- End of figure legend 4 states "All error bars (D-H) show S.E.M." - Figure 4D shows microscopy images and doesn't include error bars - Please check that all in-text reference to figures are correct.
- Figure 5A and 5B: It would be clearer to include temperature labels (@ 18oC or @ 37oC) below the

time point label to be consistent with figure 3A and 3B.

Referee #3:

The manuscript by Dr. Fischl and colleagues addresses the impact of induced cold stress on the chromatin organization in human cardiomyocytes. Furthermore, the study explores the connection of this effect to the regulation of the cell-autonomous molecular clocks. The data suggest that cold exposure leads to the nuclear accumulation of several clock components, with *Reverba* being the most affected. Of note, exposure of the AC16 cardiomyocytes to the cold stress at 28C, 18C and 8C led to the temperature specific transcription modifications in the cellular nuclei and cytoplasm. These alterations were non-linear, with the nuclear signatures at 8C being surprisingly the mildest, i.e. closest to those observed at 37C controls. Interestingly, *CIRBP* transcript that was upregulated at 28C as one would expect, did not exhibit changes at more extreme lower temperatures. Strikingly, the transcripts upregulated at 18C were retained in the nucleus and were released to cytosol upon rewarming back to 37C, including the core-clock components of the negative limb (*CRYs* and *Pers*) and *Reverba*. The latter exhibited the most dramatic changes of its levels. Subsequent experiments with U2OS cells expression *Per2::luciferase* reporter and expressing or lacking *Reverba* highlighted the role of the core-clock repressors in the cell response to the extreme cold stress and in resetting the clockwork upon rewarming. Moreover, these experiments revealed that cold induced activation of *Reverba* was sufficient to reset the phase of cellular oscillators.

In summary, this is a well-designed and rigorously conducted study, the data is presented clearly and concisely, and followed by a meaningful discussion. The reported data lead to important conclusions both scientifically and potentially clinically.

I have only few minor suggestions for this already refined work:

1. In the Abstract, in case the space allows, I would specify what kind of "human cells" the authors are working with.
2. For the experiments with U2OS/*Per2::luc* *Reverba*KO cells - the authors thoroughly explain the deletion protocol that leads to the functional disruption of the gene. Can the authors please clarify how did they confirm that *Reverba* function has been disrupted?
3. The authors report an intriguing finding that most of the transcriptional changes are observed at 18C, whereas nuclear transcriptome exhibited relatively mild alterations at extreme temperature of 8C. How do the authors explain this interesting observation?
4. Discussion p. 15 line 6: there are unnecessary parentheses appearing between the references by Ou et al and Jackson and Kochanek.

Author's response to the comments received from the reviewers

Author's response:

We thank all the reviewers for their constructive assessment and insightful comments on our paper. We have addressed all the points raised in a revised manuscript and in the point by point responses below. We have added information regarding the statistical analyses and provided additional data including: a new experiment addressing the survival rate of cells exposed to 8°C; data from an additional sequencing time point to analyse the changes in core clock gene transcript levels in the nucleus and cytoplasm after rewarming following a 5h exposure to 18°C; data that confirm disruption of *REV-ERB α* expression in the *REV-ERB α* knockout cell line. We have addressed all other points raised by the referees by changing the text in the manuscript as requested. We feel that together these changes and the additional information we have included have greatly improved our manuscript and strengthened the conclusions made and we are grateful to the referees for that.

Summary of the changes made to figures:

- 1) Fig. 1A: Replaced panels showing the DAPI stain and chromatin segmentation at 8°C with those from a stack that is more representative of the chromatin compaction at this temperature.
- 2) Fig. 1B and EV1D-E: Addition of analyses testing the significance of differences in the mean nuclear volume and chromatin volume:nuclear volume.
- 3) Fig.3 and EV3: Addition of axis labels and molecular weight markers.
- 4) Fig. EV4A: RNA-FISH data probing for *REV-ERB α* transcripts in the *REV-ERB α* KO U2OS cell line exposed to 18°C for 24h, confirming disruption of *REV-ERB α* expression.
- 5) Fig. 5: and EV4: Addition of labels clarifying the time periods during which the cells are at 37°C or 18°C.
- 6) Appendix Fig. S1: Analysis of the changes in core clock gene transcript levels in the nucleus and cytoplasm after rewarming following a 5h exposure to 18°C.
- 7) Appendix Fig. S3: Assessment of AC16 cell survival/viability after exposure to low temperatures.

Responses to specific points raised by individual referees:**Referee #1**

1 -> General Comments

This is an interesting study on the effects of sub-physiological temperature on circadian gene expression that have hitherto not yet been investigated. A question that remains open is how a surge of *REV-ERB α* accumulation can synchronize clock gene expression without affecting *CLOCK* and *BMAL1* expression. *REV-ERB α* has been thought to modulate the expression of positive limb members (i.e.), however, the authors did not observe temperature-dependent changes in *CLOCK* and *BMAL1* mRNAs. It is thus unclear how the release of *REV-ERB α* mRNA from the nucleus can quickly synchronize the different phases established by Dexamethason (see below).

In the Introduction and Discussion sections the description of the Transcriptional/Translational Feedback Loops (TTFLs) underlying circadian rhythm generation is very confusing. As it's written it gives the impression that CRYs, PERs and REV-ERBs belong to the same functional class of "repressor/corepressor" genes encoding proteins of the negative limbs of the TTFLs. According to current belief there are two coupled feedback loops driving the molecular oscillator. In the primary TTFL CRYs and PERs are assembled in large co-repressosome complexes that annul the activity of CLOCK-BMAL1 complexes. In the secondary TTFL, REV-ERB repressors cyclically inhibit the transactivation of the BMAL1 and CLOCK genes through competitive binding with ROR activators to ROREs. Since the rhythmic expression of REV-ERBs is under control of the primary TTFL, the primary and secondary TTFLs are tightly coupled. As mentioned above, the results obtained by the authors with regard to the phase-setting role of REV-ERB α in the cooling-rewarming protocol is difficult to interpret on the basis of the "coupled TTFL model". Nonetheless, the data presented in Figure 5B are rather compelling, and this reviewer thus considers likely that REV-ERB α acts through a less conventional mechanism, perhaps by binding directly to PER2 (Schmutz et al., 2010). I am not asking the authors to figure out the detailed mechanism through which Rev-ERB α synchronizes circadian clocks upon rewarming, as this would take years. However, this issue should definitely be discussed in greater detail.

We apologise for not clearly distinguishing between the primary and secondary TTFLs. We have revised the TTFL description accordingly in the introduction and discussion in the revised manuscript.

In regard to the potential mechanisms by which the accumulation of REV-ERB α triggers phase synchronisation, we have now included a new paragraph in the discussion on page 19 in the revised manuscript.

Minor Issues

I have never seen the term "clock suppressor" standing for members of the negative limbs of the TTFLs in the scientific literature. While REV-ERBs act as repressors (in the classical sense) by directly binding to cis-acting regulatory elements (ROREs), CRYs and PERs can be considered as corepressors, since they bind to DNA-binding proteins (here CLOCK-BMAL1) rather than to DNA elements. It would be advisable to replace "clock suppressors" with "clock repressors/corepressors" or "negative limb members".

We have now explained the distinction between the roles of clock co-repressor genes (*PER1/2* and *CRY1/2*) and clock repressor genes (*REVERB α*) as part of two separate negative limbs of two TTFLs more clearly in the introduction and then throughout the rest of the text referred to them together as "negative limb genes", as advised.

On page 8 (Paragraph "Cold exposure shapes...") the terms "genes" and "transcripts" are confused on multiple occasions. For example "651 and 1267 genes showing significant changes in the nucleus and the cytoplasm" should read "651 and 1267 transcripts showing

significant changes in the nucleus and the cytoplasm". Please discriminate between "genes" and "transcripts" throughout the text.

We agree that more precision is needed and in response we have now changed all the relevant examples and clearly and consistently state that the transcript levels of genes are changing.

At this point it is worth noting that the numbers of genes showing differential transcript levels upon exposure to different cold conditions has now changed slightly from the initial submission. This is because we realised that there was an error in the original table that we were using for annotations in which certain gene names had been accidentally converted to dates (e.g. SEPT1 to 1-Sep) and then excluded from further analysis as they were not recognised as genes upon intersection with genes within the RefSeq database. This error has now been corrected and has changed the significance values calculated by DESeq for a small number of genes, resulting in different numbers of genes showing significantly different transcript levels. This has not affected any of the other analyses or conclusions in any other way.

On page 18 the sentence "However, both of these pathways cannot account for the observed phase changes..." should read "However, neither of these pathways can account for the observed phase changes....".

We have now changed this.

Referee #2

Major comments

Figure 1

· Page 6 line 17/18 states "...compaction and largest increase in IC occurs at 18{degree sign}C, not 8{degree sign}C...". The chromatin landscape at 8oc looks most similar to 37oc in Fig 1A, and not like the other cooled samples. Can the authors expand on why incubating cells at 8oc does not appear to have the same effect as cooling at 28oC or 18oC?

We agree that the image that we had used to represent the compaction observed for cells exposed to 8°C gave the impression of a similar level of compaction to that seen at 37°C. We have now changed that image to that from a different z-stack of the same nucleus, which shows a similar level of compaction to that observed at 28°C. All stacks can be viewed in the movie files. The boxplots in Fig. 1B and EV1E show the distribution of ratios of chromatin volume to nuclear volume for all cells imaged for each condition. These show that, while there is some variation between cells, in the 28°C and 8°C condition cells show on average significantly more compaction than at 37°C but significantly less than at 18°C.

The effects on chromatin compaction could be different because the level of calcium influx, that we propose as a possible mechanism for chromatin compaction, may differ between the temperatures. It can be expected that both membrane fluidity and channel responses are different between the different temperatures. We have added a sentence explaining this possibility in the discussion on page 16 of the revised manuscript:

“As both the calcium channel activity and membrane fluidity (Al-Fageeh & Smales, 2006) vary at different temperatures, the calcium release is likely to differ at the three temperatures, resulting in the observed differences in the degree of compaction.”

· Page 6 line 18/19 states "...cold-induced compaction to chromatin is reversed (Fig. 1B)". However, fig 1B only shows partial reversal of compaction when incubating 18oC cells at 37oC for 2 hours. Moreover, the y-axis scale suggests that these are very moderate changes. Therefore, can the authors include statistical analysis to confirm these observations.

We have now included statistics from carrying out Mann-Whitney tests showing that upon rewarming cells for 2h there is significantly less compaction than at 18°C. There is however still significantly more compaction than at 37°C. We have therefore amended the text to clarify that the compaction to chromatin upon rewarming shows reversion back towards the levels seen in cells before cooling.

See page 6: “Upon rewarming back to 37°C, this cold-induced compaction to chromatin reverses back towards levels seen in cells before cooling (Fig. 1B)”

· What is the viability of cells grown at 8oC for 24 hours? Can the authors include analysis of cell death in each of these different cooling conditions to show that cells are not dying upon cooling and rewarming.

We have carried out viability assays for AC16 cells exposed to 28°C, 18°C and 8°C for 24h and then back up from each of these temperatures to 37°C for 24h. Whilst at both 18°C and 8°C temperatures some cells are no longer adherent and are lost upon aspiration of the growth media (~60 % of cells remain adherent after 24h at 8°C and ~80 % of cells remain adherent after 24h at 18°C). Those that remain are largely viable as shown by the number of cells approximately doubling after 24h back at 37°C, which is similar to the doubling time of AC16 cells kept at 37°C. It is important to note that only these adherent cells are included in sequencing and imaging experiments. Cells at 28°C continue to proliferate at the lower temperature but at a slower rate.

Figure 3

· Page 9 line 20 states "...rewarming cells back from 18{degree sign}C to 37{degree sign}C leads to a rapid readjustment of the nuclear-accumulated transcripts back to levels seen before cooling (Fig. 3A, top panel)". Can the authors include statistical analysis to support this conclusion.

We have included p-values from paired t-tests to show the significance of any difference in standardized RNA levels between key time points **in the source data file** for each of the standardized RNA level boxplots. We have not included these p-values in the figure as we do not think this adds value. This is because many genes are involved for each comparison, so many of the small differences observed between time points are significant. Therefore, upon rewarming from 18°C back to 37°C for 2h, nuclear-accumulated transcripts are still

significantly elevated relative to their level at 37°C, but these levels are much closer to their levels at 37°C than at 18°C for 24h.

We have amended the text to clarify this.

See page 9: “Interestingly, rewarming cells back from 18°C to 37°C leads to a rapid readjustment of the nuclear-accumulated transcripts back to levels close to those seen before cooling (Fig. 3A, top panel).”

· Figure 3D: It is important to show that the tagging of REV-ERBa doesn't perturb its function or mRNA localisation upon cooling and rewarming. It would also be interesting to know if the translation of REV-REBa correlates with global protein synthesis rates.

The Western blots match what we expect to see based on the RNA-seq and RNA-FISH results. When *REV-ERBa* RNA is elevated in the nucleus but not in the cytoplasm then there is no increase in the tagged-REV-ERBa Western blot signal (e.g. after 5h at 18°C). When *REV-ERBa* RNA is elevated in the cytoplasm (e.g. after 24h at 18°C or upon rewarming after 5h at 18°C) then the tagged-REV-ERBa Western blot signal is increased. This suggests that the tag does not affect the localisation of the *REV-ERBa* RNA. While we cannot be sure that the tag does not disrupt REV-ERBa function, the tagged strain was not used in any other experiments other than for these Western blots and so any disruption to function is of less importance. The change in the level of REV-ERBa translation after 24h at 18°C or upon rewarming after 5h at 18°C is greater than that of the rate of global protein synthesis based on the Ponceau stain and TP53 translation based on the control Western blot.

· Text on page 11 line 9/10 states "...nuclear-retained transcripts surge into the cytoplasm after rewarming to 37{degree sign}C where they are translated (Fig. 3D, S3H)". However, fig 3D and S3H only shows data for REV-ERBa and not PER1, CRY1 or CRY2. Please amend the text or include examples of other nuclear transcripts to support this text.

We agree and have made this point clearer by amending the text accordingly: “The nuclear-retained transcripts surge into the cytoplasm after rewarming to 37°C where, as shown for *REV-ERBa*, they can be translated (Fig. 3D, EV3H).”

· Text on page 11 line 18/19 states "...we employed RNA-FISH using probes targeting transcripts for the most highly upregulated and earliest responding gene, REV-ERBa, the later responding and nuclear-restricted gene, CRY2, and the unchanged control gene, TP53 (Fig. S3F)". Fig S3F shows ARNTL, CLOCK and TP53. This is very confusing - please amend.

We apologise for the confusion caused and have re-written this section to clarify this issue on page 11 as follows:

“To corroborate the observed RNA-Seq expression patterns of clock genes, we employed RNA-FISH using probes targeting the following transcripts: *REV-ERBa*, showing the highest upregulation and earliest response (Fig. 3B); *CRY2*, showing a later response and nuclear-restricted upregulation (Fig. 3B); *TP53*, showing no changes and acting as a control (Fig. EV3F).”

Figure 4

· Figure 4B and 4F - The representative images in 4B (particularly 18oC 24h, 37oC 2h) do not support the quantified data in 2F. This makes it difficult to conclude: "... after 24h, CRY2 transcripts accumulate in the nucleus and are released into the cytoplasm after rewarming". Please include the N value of replicates used to generate this quantification.

We respectfully disagree with this comment. The representative images show an increase in CRY2 transcript nuclear dots at the 18°C 24h time point and an increase in cytoplasmic dots at the 18°C 24h, 37°C 2h time point. These dots have been counted for many images of cells exposed to each condition (see source data file and Table S7). The number of images and replicates for each condition was stated in the methods. We have now included the following statement in the figure legend: "Number of images and actual p-values are presented in Table S7 and the source data file" and included the relevant information in the source data file. The counts for each image are also included in the source data file. All the RNA FISH images have also been deposited in the Image Data Resource (<https://idr.openmicroscopy.org>) under accession number idr0089. The codes used to automate the counting of dots per nuclear and cytoplasmic area have also been deposited on Github (<https://github.com/hjf34/Cold>).

· Page 13 line 11/12 states "...rewarming following exposure to 18{degree sign}C for 5h only triggers a cytoplasmic pulse of increased REV-ERB α without affecting PER1/2 or CRY1/2 levels (Fig. 3-4)". How can the authors conclude this when the data for CRY1/2 or PER1/2 at the 5 hr time point are not included in figure 4? The authors must include the 5 hr time for at least CRY2 to make this conclusion. Moreover, figure 3B only shows the localisation of RNA for rewarming after 24 hr 18oC and not 5 hours. Please include the data for 5 hours to support this statement.

In the original submission, we showed by RNA-seq that, out of the core circadian clock genes, *REV-ERB α* was the only one to show elevated transcript levels in the nucleus after 5h and by RNA-FISH and western blot that rewarming following this short exposure triggered a cytoplasmic pulse of increased *REV-ERB α* transcripts and an increase in *REV-ERB α* protein levels. We assumed that, because none of the other core circadian clock gene transcripts showed elevated levels in the nucleus, we would see no change in their cytoplasmic levels upon rewarming. We agree with the reviewer that we could not be sure of this, so we have now carried out two RNA-seq replicates for nuclear and cytoplasmic fractions of AC16 cells exposed to 18°C for 5h and then returned to 37°C for 90min. These show an increase in cytoplasmic *REV-ERB α* transcript levels as the nuclear levels decrease, as expected; however, there is also an increase in the cytoplasmic and nuclear levels of *PER2* transcripts but not *CRY1/2* or *PER1* transcripts (see Fig. S1). We also observe this increase in *PER2* transcript levels upon rewarming following exposure to 18°C for 24h (Fig. 3B). We have now amended the text to account for this new observation.

See page 13: "Rewarming following exposure to 18°C for 5h triggers a cytoplasmic pulse of increased *REV-ERB α* without affecting *PER1* or *CRY1/2* levels (Fig. 3-4, S1). We could therefore use this to assess the effect of this cold-induced cytoplasmic surge of *REV-ERB α*

on the circadian rhythm in isolation from the other clock negative limb genes, except for *PER2*, which shows increased transcript levels after rewarming (Fig. S1).”

· The authors show that the localisation of REV-ERBa in the cytoplasm following 18oC 24 hr and rewarming 2 hr differs in AC16 and U2OS cells (Fig 4E and 4H - purple bar in cytoplasmic area), suggesting that REV-ERBa is retained/maintained in the cytoplasm following rewarming in U2OS cells but not AC16 cells. Could the authors provide an explanation as to why this may be the case?

Analysis of the RNA-seq data for U2OS cells shows that the pattern of expression changes of *REV-ERBa* transcripts relative to cells kept at 37°C in both the nucleus and cytoplasm is very similar to that of AC16 cells for all the time points for which we have data in both cell lines (18°C for 5h, 18°C for 24h, 18°C 24h and then back at 37°C for 2h) (compare Fig. 3B to Fig. EV3G). RNA-FISH, unlike 3' end RNA-seq, does not require transcripts to be polyadenylated in order for their detection. The RNA-FISH data are therefore likely to detect some transcripts that are in the process of being degraded, which have had their polyA tails removed or shortened. We suspect that the degradation rate for the *REV-ERBa* transcripts in the cytoplasm upon rewarming may be slightly slower in U2OS cells than in AC16 cells, leading to slightly longer maintenance of these transcripts in the cytoplasm, particularly of transcripts in the process of being degraded that can be detected by RNA-FISH but not RNA-seq, thus giving the higher cytoplasmic RNA-FISH signal in U2OS cells after two hours back at 37°C, following 24h at 18°C. We do not believe that this elevated level of cytoplasmic transcript signal will last for long given that the levels of nuclear transcripts are reduced by this time point. Furthermore, transcripts without poly A tails will not be translated and so will not affect REV-ERBa protein levels.

Figure 5

With the differences observed across cell lines at the 24 hour time point, can the authors justify why all subsequent experiments at the 24 hr cooling time point in figure 5 were only carried out in U2OS and not in AC16 cells? The experiments at 24 hours in figure 5 should incorporate AC16 cells to ensure that the effects observed at the later time points are not cell line specific, and this would also increase the therapeutic relevance of the study.

As explained above we do not think that there are major differences between the patterns of expression changes shown by the two cell lines after two hours back at 37°C, following 24h at 18°C. In fact, we show that the expression changes are remarkably similar between the two cell lines for all circadian clock gene transcripts across all time points when measured using RNA-seq. We have also demonstrated that after 24h at 18°C, REV-ERBa is not required for the effects observed on the phase or amplitude of the circadian rhythm (Fig. 5A). Therefore, any difference in REV-ERBa levels between the two cell lines at this time point are not of great importance.

The U2OS *PER2::LUC* cell line was used to carry out the investigations into the circadian rhythm as it is a well-established cell line for such studies (e.g. Jagannath et al., 2013). We believe that as we have shown that the expression patterns for the circadian clock gene transcripts are sufficiently similar between the two cell lines, it is adequate to assume that the

circadian rhythms of these two cell lines will be affected in a similar manner to exposure to 18°C.

Minor Comments

- Add N values to all figure legends.

The N values for each experiment have been included in the methods or in the source data file. For cases where inclusion of the N value for each condition would lead to overly lengthy figure legends (e.g. Fig. 4) we have included a statement in the legend explaining where to find the N values.

- Figure 1B - Add missing value to axis (0.60).

This axis has now been extended to include this value.

- Refer to the figures in order and provide the figure reference at the appropriate place in the text.

We have addressed this as well as possible

- In figures where multiple cell lines are used, please state the cell line used in the figure legend.

For all figures where there are multiple cell lines it is now clearly stated which cell line has been used.

- Axis titles missing in figures 3B, 3D

Axis titles have now been included in these figures

- Please add protein size marker label to ponceau stain in figure 3D and S3H

Protein size markers have now been included in these figures.

- Figure 3D: It is not clear if these are cytoplasmic cell extracts. Please add more detail to the figure legend.

These are whole cell extracts. This was previously explained in methods and we have now included this information in the figure legend.

- @ 37°C is missing from figure S3A

This has now been included

- End of figure legend 4 states "All error bars (D-H) show S.E.M." - Figure 4D shows microscopy images and doesn't include error bars - Please check that all in-text reference to figures are correct.

This has now been changed to read E-H. We have checked that all other in-text references to the figures are correct.

- Figure 5A and 5B: It would be clearer to include temperature labels (@ 18°C or @ 37°C) below the time point label to be consistent with figure 3A and 3B.

We have now included labels in this figure to clearly indicate in which time period the cells are at 37°C and at 18°C.

Referee #3

I have only few minor suggestions for this already refined work:

1. In the Abstract, in case the space allows, I would specify what kind of "human cells" the authors are working with.

With all due respect we would rather not specify the cell lines in the abstract as it may imply that the findings are cell type-specific and our results suggest that this is not the case.

2. For the experiments with U2OS/Per2::luc ReverbaKO cells - the authors thoroughly explain the deletion protocol that leads to the functional disruption of the gene. Can the authors please clarify how did they confirm that Reverba function has been disrupted?

Thank you for highlighting this point. We have now included RNA-FISH data probing for *REV-ERB α* transcripts in the U2OS *REV-ERB α* KO cell line after exposing these cells to 18°C for 24h (Fig. EV4A). This shows a large significant reduction in *REV-ERB α* transcript levels relative to the WT cell line showing that the deletion has disrupted *REV-ERB α* expression. Some signal for *REV-ERB α* transcripts remains in the KO cells as only the promoter and first exon of *REV-ERB α* has been deleted. This deletion of the first exon should also cause a frame shift in the protein, which is likely to also disrupt its function. We have amended the text in the methods to clarify that we have not directly tested whether the function has been disrupted.

See page 21: "This leads to a frame shift of subsequent exons, which is likely to disrupt REV-ERB α function."

3. The authors report an intriguing finding that most of the transcriptional changes are observed at 18C, whereas nuclear transcriptome exhibited relatively mild alterations at extreme temperature of 8C. How do the authors explain this interesting observation?

In the discussion section on pages 16-17 we provide a possible explanation for this observation. We suggest that a drop in temperature to 8°C may deplete ATP to levels, which will impact on RNA polymerase II processivity. We have now added that enzyme activity at 8°C and 18°C is also likely to differ which contributes to the observed difference in the transcriptional response at the two temperatures. We suggest that at 18°C enzyme RNA polymerase II enzyme activity still sufficient to enable the gradual changes to the nuclear transcriptome observed over the 24h time period.

Accordingly, we have expanded the relevant section in the discussion and added the following lines on pages 16/17 of the revised manuscript:

"Furthermore, it is likely that RNA polymerase II enzyme activity is much more severely compromised at 8°C compared to 18°C where a transcriptional response is still maintained.

However, it is worth noting that the transcriptional response at 18°C is gradual and requires exposure for 24h to affect the transcript levels of 1000 genes (EV2F)”

4. Discussion p. 15 line 6: there are unnecessary parentheses appearing between the references by Ou et al and Jackson and Kochanek.

Thank you, we have changed this mistake in the revised version.

Dr. Andre Furger
University of Oxford
Biochemistry
South Parks Road
Oxford OX1 3RE
United Kingdom

19th Aug 2020

Re: EMBOJ-2020-105604R

Cold-induced chromatin compaction and nuclear retention of clock mRNAs resets the circadian rhythm

Dear Andre,

Thank you for submitting your revised manuscript, we have now received the reports from the three initial referees (see comments below). I am pleased to say that they overall find that their comments have been satisfactorily addressed and now support publication. Referee #2 raises a minor issue regarding the figure legends, which you will see is in part also brought up by our data editors and should be resolved in the final revised version. In addition to this, I would like to ask you to also address a number of editorial issues that are listed in detail below. Please make any changes to the manuscript text in the attached document only using the "track changes" option. Once these remaining issues are resolved, we will be happy to formally accept the manuscript for publication.

Thank you again for giving us the chance to consider your manuscript for The EMBO Journal. I look forward to receiving your final revision. Please feel free to contact me if you have further questions regarding the revision or any of the specific points listed below.

Kind regards,

Stefanie

Stefanie Boehm
Editor
The EMBO Journal

Referee #1:

As stated in my review of the original version, this is an interesting study with unexpected findings that are relevant for a wide readership. In their revised version the authors have now satisfied all of my queries by editorial changes.

Referee #2:

This is an interesting piece of work and will be of value to the field. The authors have addressed the majority of my concerns. However, N values to provide significance to the data really need to be in the figures legends and not elsewhere in the document that is hard to access, even if this makes the legends a little long.

Referee #3:

The authors have conducted a thorough revision of their work, and have addressed all the raised commentaries in a satisfactory manner.

Dr. Andre Furger
University of Oxford
Biochemistry
South Parks Road
Oxford OX1 3RE
United Kingdom

29th Aug 2020

Re: EMBOJ-2020-105604R1

Cold-induced chromatin compaction and nuclear retention of clock mRNAs resets the circadian rhythm

Dear Andre,

Thank you again for submitting the final revised version of your manuscript. I am pleased to inform you that we have now accepted it for publication in The EMBO Journal.

Your article will be processed for publication in The EMBO Journal by EMBO Press and Wiley, who will contact you with further information regarding production/publication procedures and license requirements.

Should you be planning a Press Release on your article, please get in contact with embojournal@wiley.com as early as possible, in order to coordinate publication and release dates.

Congratulations on your successful publication, and thank you again for this contribution to The EMBO Journal! Please continue to consider EMBO Journal for your work in the future.

Kind regards,

Stefanie

Stefanie Boehm
Editor
The EMBO Journal

-

Corresponding Author Name: Andre Furger
 Journal Submitted to: THE EMBO JOURNAL
 Manuscript Number: EMBOJ-2020-105604R1